# Magnesium in subaqueous speleothems as a potential palaeotemperature proxy

Russell Drysdale [1,2✉], Isabelle Couchoud[1,2], Giovanni Zanchetta [3], Ilaria Isola [4], Eleonora Regattieri[5], John Hellstrom[6], Aline Govin [7], Polychronis C. Tzedakis [8], Trevor Ireland [9], Ellen Corrick[1], Alan Greig[6], Henri Wong[10], Leonardo Piccini [11], Peter Holden[9] & Jon Woodhead [6]

Few palaeoclimate archives beyond the polar regions preserve continuous and datable palaeotemperature proxy time series over multiple glacial-interglacial cycles. This hampers efforts to develop a more coherent picture of global patterns of past temperatures. Here we show that Mg concentrations in a subaqueous speleothem from an Italian cave track regional sea-surface temperatures over the last 350,000 years. The Mg shows higher values during warm climate intervals and converse patterns during cold climate stages. In contrast to previous studies, this implicates temperature, not rainfall, as the principal driver of Mg variability. The depositional setting of the speleothem gives rise to Mg partition coefficients that are more temperature dependent than other calcites, enabling the effect of temperature change on Mg partitioning to greatly exceed the effects of changes in source-water Mg/Ca. Subaqueous speleothems from similar deep-cave environments should be capable of providing palaeotemperature information over multiple glacial-interglacial cycles.

[1] School of Geography, The University of Melbourne, Parkville 3010 VIC, Australia. [2] Laboratoire EDYTEM, UMR CNRS 5204, Université Savoie Mont Blanc, 73376 Le Bourget-du-Lac cedex, France. [3] Dipartimento di Scienze delle Terra and CIRSEC, University of Pisa, 56126 Pisa, Italy. [4] Istituto Nazionale di Geofisica e Vulcanologia, 56126 Pisa, Italy. [5] Istituto di Geoscienze e Georisorse, IGG-CNR, Via Moruzzi 1, 56126 Pisa, Italy. [6] School of Earth Sciences, The University of Melbourne, Parkville 3010 VIC, Australia. [7] LSCE-IPSL (CEA-CNRS-UVSQ), Paris-Saclay University, 91190 Gif-sur Yvette, France. [8] Environmental Change Research Centre, Department of Geography, University College London, London WC1E 6BT, UK. [9] Research School of Earth Sciences, The Australian National University, Canberra 2600 ACT, Australia. [10] Australian Nuclear Science and Technology Organisation, Lucas Heights, NSW 2234, Australia. [11] Dipartimento di Scienze delle Terra, Universita degli Studi di Firenze, Via la Pira 4, 50121 Firenze, Italy. ✉email: rnd@unimelb.edu.au

Records of past temperature change during Quaternary glacial-interglacial (G–IG) cycles are derived primarily from polar ice cores[1,2] and ocean sediments[3]. Beyond the polar regions, long terrestrial records of palaeotemperatures are sparse, because most archives lack chemical, biological, and/or physical properties that behave consistently as a palaeothermometer. This constitutes a major obstacle to unravelling geographic patterns of past temperature change.

Magnesium-to-calcium ratios (Mg/Ca) in carbonates are widely used in the marine sciences to study surface- and deep-ocean temperatures[4–7]. The partitioning of Mg into marine calcium carbonates, such as the calcite tests of planktonic and benthic foraminifera, is dependent primarily upon source-water Mg/Ca and mineralization temperature[8], with smaller additional control from pH, salinity and vital effects of individual species[6,9]. Open-ocean salinity variations are relatively small over G–IG time scales[10] due to the size of the ocean reservoir. Therefore, changes in seawater temperature drive most of the Mg/Ca variation in marine carbonates[11]. In terrestrial carbonates (e.g., speleothems and lake sediments), on the other hand, variations in source-water Mg/Ca are usually large, significantly overriding the effects of temperature on partitioning[12,13]. This limits the use of Mg/Ca as a terrestrial palaeothermometer.

Speleothems have become widely used in palaeoclimatology over the last two decades due to their sensitivity to climate and environmental change[14], and their potential for establishing the timing of climate events with high precision[15,16]. The temperature deep inside a cave system from where speleothems are usually sampled is assumed to reflect mean annual external air temperature[14], yet the ability of speleothems to preserve thermal histories is impeded by the multiplicity of factors that cause variations in temperature-sensitive geochemical tracers[17,18]. Variations in speleothem Mg/Ca have been shown to reflect mainly changes in site hydrology[18,19] rather than temperature. Periods of reduced recharge can lead to partial dewatering of fractures and longer water–rock interaction times. This can cause prior calcite (or aragonite) precipitation (PCP) and/or incongruent dissolution of dolomite (IDD)[19], both of which can raise percolation-water Mg/Ca by 50% or more around mean values[20,21]. These changes override the effect of temperature on the partitioning of Mg from the source water to the speleothem and produce the familiar pattern of higher (lower) speleothem Mg/Ca during periods of low (high) recharge[22–28]. These findings

have come principally from studies of stalagmites and flowstones, the most common speleothem types used in climate reconstructions. Few palaeoclimate studies have investigated the utility of subaqueous speleothems in this regard, in spite of their ability to host long, continuous palaeoclimate records[29–32], and despite the fact that the pools and lakes in which they grow are common features of cave systems.

Here we investigate the potential of using Mg palaeothermometry in a subaqueous speleothem (CD3) from Corchia Cave, Italy[30,33]. The cave is well located to conduct such a study: it is situated close to the North Atlantic, whose ocean heat transport modulates the climate of Western Europe and the Mediterranean[34]. Changes in regional sea-surface temperatures (SSTs) should therefore trigger concomitant changes in air temperatures above Corchia Cave. These, in turn, will alter cave-interior temperatures. To test this, we measured stable isotope ratios ($\delta^{18}O$ and $\delta^{13}C$) and Mg concentrations covering the last ~350,000 years (~350 ka) in a core drilled from CD3, and compared the results to a composite ocean-sediment record from the Iberian margin (sites MD01-2443 and MD01-2444)[35–40]. We first determined how well the speleothem Mg tracks regional SSTs over multiple G–IG cycles. We then measured $\delta^{18}O$ and Mg on the same speleothem at higher resolution across Termination II (T-II: ~136–129 ka)[16,41], an interval well-captured in the same ocean-sediment cores, to make similar Mg-SST comparisons. There is clear correspondence between Mg and SST at the orbital scale, but across T-II the structural agreement is very strong, particularly in that Mg captures the sharp rise in SST at the end of the Heinrich Event 11 (H11) stadial. We also find that Mg responds more consistently to temperature than either speleothem $\delta^{18}O$ or $\delta^{13}C$. This unusual temperature sensitivity of Mg can be attributed to the depositional setting of CD3, where very slow accretion of calcite occurs from a low-saturation, low-ionic-strength solution. We conclude that high-resolution Mg profiles from this and similar speleothems have excellent potential for providing rare terrestrial palaeotemperature records, and for tracking major changes in regional SSTs. Besides providing much-needed temperature data, this may ultimately help anchor ocean-sediment records in radiometric time through Mg-SST synchronization.

## Results

**Cave setting**. Antro del Corchia (Italy; Fig. 1) is an extensive cave system developed in Mesozoic marbles, dolomitic marbles and dolomites of the Alpi Apuane karst, northern Tuscany (Fig. 1)[33,42]. The cave contains a large, well-decorated chamber (Galleria delle Stalattiti; 835 m.a.s.l.) that hosts several pools[42,43], the wetted perimeters of which are blanketed by subaqueous calcite[30,33]. CD3 is a prominent, dome-like calcite mound that has grown directly over the dolomite bedrock substratum of the largest pool, Laghetto Basso[30,33,44]. A 27 cm core (CD3-1) was drilled from the mound in 2007 and was prepared for trace-element and stable isotope analyses, and uranium–thorium (U-Th) dating (see 'Methods', Supplementary Fig. 1 and Supplementary Data 1). A preliminary age profile has revealed slow but continuous calcite accretion (mean: ~0.3 mm kyr$^{-1}$) since ~970 ka[16,30].

The estimated mean altitude of recharge precipitation (~2500 mm yr$^{-1}$) reaching the Galleria delle Stalattiti is ~1500 m.a.s.l. based on the mean pool-water $\delta^{18}O$ ($-7.4 \pm 0.1$‰, 1 SEM)[44] and the local altitude-$\delta^{18}O$ relationship established from non-karstic spring waters across the Alpi Apuane[43,45] (Supplementary Fig. 2). Before reaching the pool, infiltration waters traverse mostly dolomitic host rock and are of low ionic strength, with relatively

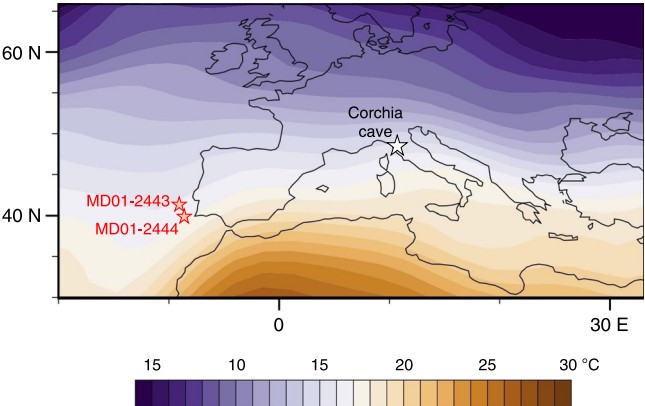

**Fig. 1 Mean annual regional air temperatures and study site.** Location of Corchia Cave (Italy) and the two Iberian margin ocean-coring sites MD01-2443 and MD01-2444. The coloured contouring is the mean annual air temperature field at 1000 mbar for the period 1981–2010. Image is provided by the NOAA-ESRL Physical Sciences Laboratory, Boulder, Colorado, from their Web site at http://www.esrl.noaa.gov/.

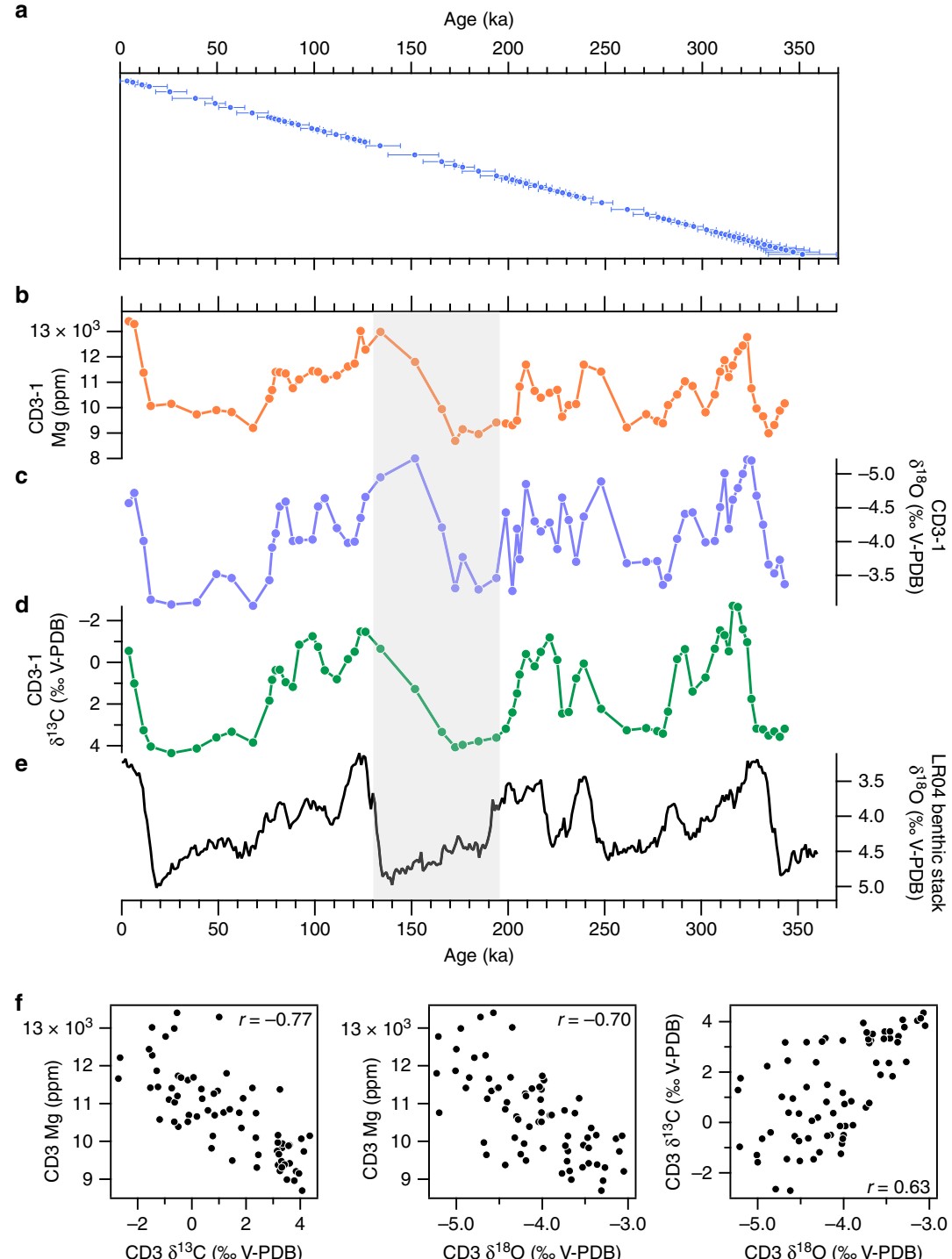

**Fig. 2 CD3-1 data spanning the last 350,000 years. a** The blue circles show the distribution across time of 115 radiometric ages (see Supplementary Data 1) on CD3-1 (see Supplementary Fig. 1 for the age-depth model and the age uncertainty time series). The error bars are $2\sigma$ uncertainties based on the U-Th age calculations. **b–d** Mg, $\delta^{18}O$ and $\delta^{13}C$ series from CD3-1 sampled at 1 mm resolution (see 'Methods'). **e** The Lisiecki and Raymo[51] LR04 benthic $\delta^{18}O$ stack. The grey vertical band is the MIS 6-to-Termination II (~195–130 ka) interval where the greatest age discrepancy occurs between the CD3-1 series and the benthic $\delta^{18}O$ (see main text for discussion). **f** Scatterplots and Pearson's $r$ correlation coefficients for CD3-1 data shown in **b–d**. All $r$-values are statistically significant at $p < 0.05$.

invariant chemical and isotopic composition[33,42,44]. The mean air temperature of the gallery is $8.4 \pm 0.3$ °C (1 SD), about 3 °C lower than the external mean annual air temperature (MAAT; 11.8 °C) measured at a similar elevation near the tourist entrance to the cave (860 m.a.s.l.)[42]. Based on ~monthly measurements between May 2009 and March 2012, the mean water temperature of

Laghetto Basso is slightly cooler than that of the Galleria delle Stalattiti air temperature ($7.9 \pm 0.2$ °C)[33]. Discontinuous carbon dioxide monitoring conducted in the gallery between September 1997 and April 1999 revealed concentrations ranging between 387 and 1324 p.p.m.v. (mean $628 \pm 198$ p.p.m.v.; based on ~200 days of data)[46].

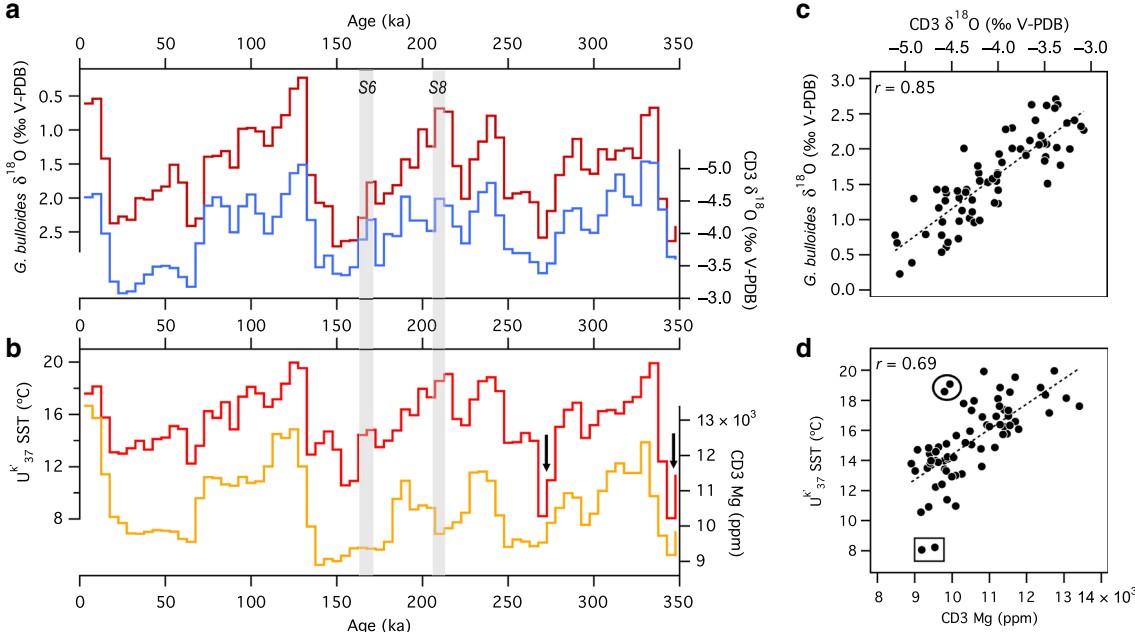

**Fig. 3 CD3-1 and ocean-core data for the last 350,000 years. a** Synchronization of the CD3-1 $\delta^{18}O$ series (blue) to the planktic foraminiferal $\delta^{18}O$ from *G. bulloides* (crimson) from sites MD01-2443/2444 (see 'Methods' and Supplementary Fig. 3). **b** CD3-1 Mg (orange) compared to the alkenone $U^{k'}_{37}$ sea-surface temperatures (SST) from sites MD01-2443/2444 (red). The Mg captures much of the orbital-scale variation in the SST. Grey vertical bands are the position of sapropel events S6 and S8, whose ages are based on the adjusted CD3-1 chronology. **c, d** Correlations arising directly from the synchronization between CD3-1 $\delta^{18}O$ and planktic $\delta^{18}O$ **a**, and between CD3-1 Mg and SST shown in **b**. Black circle and square in **d** highlight outlier points discussed in the text. Black arrows in **b** indicate intervals where exceptionally low SSTs are not matched by correspondingly low Mg concentrations. All series have been resampled at 5 kyr resolution (see 'Methods').

**Comparison of CD3 and ocean-sediment records since 350 ka.**
Stable isotope and Mg data measured from CD3-1 display strong covariation over orbital time scales (Fig. 2). In general, the $\delta^{13}C$ in Corchia speleothems responds to changes in soil/vegetation activity above the cave, the biogenic $CO_2$ from which drives the $\delta^{13}C$ to lower values during warm climate stages[43,47,48]. Cold conditions reduce plant cover above the cave and promote erosion of the thin soils by runoff and/or cryogenic processes, leading to lower biogenic $CO_2$ input to the percolation waters and higher speleothem $\delta^{13}C$ values. Periods of lower (higher) $\delta^{13}C$, typically corresponding to interglacial/interstadial (glacial/stadial) stages, generally occur when Mg is higher (lower). The Mg is therefore high during warm climate states and vice versa. As with the $\delta^{13}C$, the speleothem $\delta^{18}O$ displays orbital-scale variations (Fig. 2). Previous work on Corchia speleothems has shown that $\delta^{18}O$ is linked to changes in regional SSTs, which affect moisture advection from the North Atlantic across the Mediterranean[43,49,50]. During interglacial/interstadial climates, the $\delta^{18}O$ is lower because warmer SSTs generate higher rainfall amounts at the cave site; the converse occurs during glacial/stadial periods. The major variations in Mg follow those in $\delta^{18}O$: the lower $\delta^{18}O$ values corresponding to warm periods largely coincide with higher Mg and vice versa. The tendency for orbital-scale variations in Mg to closely follow those from both $\delta^{18}O$ and $\delta^{13}C$ is reflected in the strong covariations shown in Fig. 2f. This is the first time that Mg in a speleothem has been shown to vary positively with temperatures.

Figure 2e shows the LR04 benthic $\delta^{18}O$ stack[51], a proxy for changes in global ice volume. The major G–IG features of the benthic $\delta^{18}O$ are reproduced in the CD3-1 record, providing supporting evidence that the speleothem is capturing orbital-scale climate changes. This is particularly compelling for $\delta^{13}C$, where there is close peak-for-peak agreement with the benthic foraminifera $\delta^{18}O$. However, age offsets are apparent. This is

most notable during the MIS 6–T-II interval (grey band, Fig. 2), where the increase in Mg and decrease in both $\delta^{18}O$ and $\delta^{13}C$ significantly lead the decrease in benthic $\delta^{18}O$. This is notwithstanding the fact that the LR04 $\delta^{18}O$ stack has age uncertainties of ±4 kyr over the last 1 Ma. Comparing CD3-1 stable isotope patterns with those for coeval stalagmites from the same cave chamber[40,49] reveals offsets as much as 30 kyr (Supplementary Fig. 3). A similar problem has been documented in speleothem CD3 for the late glacial-Holocene interval[33]. The most likely cause of these age offsets is scavenging of $^{230}Th$ from the radioactive decay of $^{234}U$ in the water column in which the speleothem grew. This scavenging process adds excess (non-authigenic) $^{230}Th$ to the calcite, leading to ages that are too old, and has been shown to affect other very slow-growing subaqueous speleothems[31,52] (see 'Methods').

To overcome the age-offset issue, we adjusted the U-Th chronology by synchronizing the CD3-1 $\delta^{18}O$ to the $\delta^{18}O$ measured on the planktic foraminifer *Globigerina bulloides* in ocean sediments from the Iberian margin (Fig. 1; see 'Methods' and Supplementary Fig. 4). Planktic $\delta^{18}O$ is a plausible first-order tuning target because, like speleothem $\delta^{18}O$, it is susceptible to changes in both mineralization temperature and source-water $\delta^{18}O$. Planktic $\delta^{18}O$ has been verified as a reliable tuning target in previous studies comparing records from Corchia and nearby ocean sediments[16,41,53]. Figure 3a shows the results of the tuning. The significant improvement in agreement between the CD3-1 and stalagmite chronologies through the MIS 6/T-II interval (Supplementary Fig. 3) lends strong support to this cave-ocean synchronization. We can now compare the CD3-1 Mg and ocean-core SSTs on a common age scale (Fig. 3b; see 'Methods').

The results show that much of the large-scale variation in SST is captured in the CD3-1 Mg (Pearson's $r = 0.69$, $p < 0.05$, df = 68: Fig. 3d). The two data points located most adrift of the line-of-best-fit correspond to the two most extreme cooling events of

the last 350 kyr recorded off the Iberian margin (at 343 ka and 268 ka: see black arrows in Fig. 3b and the points highlighted by the black box in Fig. 3d). Although these occur when Mg is low, the magnitude of the SST decrease is not captured proportionally in the CD3-1 Mg. The reason for this is unclear. Whilst the sharp SST decrease is not matched by an increase in planktic $\delta^{18}O$, as would be expected from a cooling, it does coincide with a rise in the $C_{37:4}$ freshwater biomarker[3], suggesting a Heinrich-like meltwater pulse buffered the cooling effect on planktic $\delta^{18}O$. In any case, the discrepancy may be due to the inability of CD3-1 to register excessively cold events: such events would likely cause a dramatic reduction in growth rates, resulting in a bias against the coldest temperatures, particularly given the 5 kyr resampling interval used.

Additional deviations from the Mg-SST line-of-best-fit may be attributed to 'cold sapropel' events S6 (~168 ka) and S8 (~213 ka) (grey bands, Fig. 3a, b). Sapropels are linked to tropical pluvial periods when Northern Hemisphere summer insolation intensity is high[54,55]. In the northern Mediterranean borderlands, they lead to enhanced summer aridity but intense autumn–winter precipitation[56]. Cold sapropels occur during unusually high insolation peaks in glacial times[57]. Palaeoclimate archives from western Italy preserve the regional hydroclimate response to both warm[58,59] and cold sapropel events[60], with both types producing wetter intervals. This drives $\delta^{18}O$ to lower values due to the rainfall-amount effect[43,58,60]. During cold S6 and S8, we see low values of both Mg and $\delta^{18}O$ in CD3-1 (Fig. 3a, b), particularly for S8 (see black-circled values in Fig. 3d), as would be expected if temperatures were low at a time when rainfall was relatively high. However, the Iberian-margin planktic $\delta^{18}O$ remains relatively high during these intervals due to surface ocean cooling at a time of stadial conditions. This highlights an inconsistency in aligning the Corchia $\delta^{18}O$ and Iberian-Margin planktic $\delta^{18}O$ during cold sapropels (particularly S8), because these events do not appear to affect the isotopic composition of the surface ocean waters off the Iberian margin[39].

In summary, at the orbital scale all three Corchia proxies—Mg, $\delta^{13}C$ and $\delta^{18}O$—exhibit patterns suggestive of some degree of temperature control. Periods of low $\delta^{13}C$ and $\delta^{18}O$ coincide with high Mg, indicating that Mg is responding principally to changes in temperature rather than hydrology. Following a synchronization procedure, these patterns are confirmed in a comparison between cave and ocean-sediment data. However, scatter around the lines-of-best fit in the cave data and the cave-vs.-ocean-sediment correlations suggest other processes affect the cave proxies, particularly during millennial-scale climate events such

as those associated with sapropels. For example, previous work from Corchia Cave has shown that the lowest $\delta^{13}C$ values are attained well after interglacial SST optima are reached[16,43,58], a trend attributed largely to slow rates of postglacial development of soils[48]. Thus, whilst the trend towards progressively lower $\delta^{13}C$ values generally coincides with transitions from glacial to interglacial states, slow pedogenesis limits the utility of $\delta^{13}C$ to act as a consistent temperature proxy. We now further investigate the temperature sensitivity of Mg and $\delta^{18}O$ using higher-resolution data across a glacial termination.

**CD3 and ocean-sediment records through Termination II**. A glacial termination provides an ideal time interval for investigating the reliability of Mg as a temperature proxy due to the large temperature changes occurring during these transitions. T-II (~136–129 ka)[16,41] has long been an intensive focus of study using cave[61–65] and ocean-sediment records[66,67], including records from Corchia[47,49] and the Iberian margin/western Mediterranean region[40,68–72].

Changes in Mg and $\delta^{18}O$ across T-II were explored using high-resolution, micro-analysis techniques on a section of CD3-1 (see 'Methods', Supplementary Fig. 5 and Fig. 4a). The results show that both series compare well with the lower-resolution data from drilled powder samples (Fig. 4b, c). The high-resolution Mg values are low at the base of the measured section until ca. 2.8 mm, then rise sharply to persistently high values over the upper 2 mm. The $\delta^{18}O$ values commence at intermediate levels, increase between 4 and 3 mm depth, then decrease sharply to remain low over the upper 2.8 mm of the measured section. The large decrease in $\delta^{18}O$ at ~3 mm leads the large Mg increase by a few hundred microns (Fig. 4a). This critical decoupling between Mg and $\delta^{18}O$ suggests the most prominent changes in temperature and recharge rainfall $\delta^{18}O$, respectively, across the termination were out of phase. To quantify the phasing between these major shifts in the two proxies, we synchronized the CD3-1 $\delta^{18}O$ to the recently compiled Corchia Cave stalagmite $\delta^{18}O$ stack (CCSS-18) through T-II, which is anchored by an accurate and precise U-Th chronology[40]. This bypasses the aforementioned U-Th issues with CD3-1: stalagmites are virtually immune from Th scavenging due to the short contact time between the thin film of drip water (compared to the persistent contact with a thick pool-water column) and the accreting calcite surface. In spite of the relatively large analytical uncertainties of the ion-microprobe measurements (~0.5‰, see 'Methods'), the ion-probe-derived CD3-1 $\delta^{18}O$ shows very similar changes to the conventionally measured

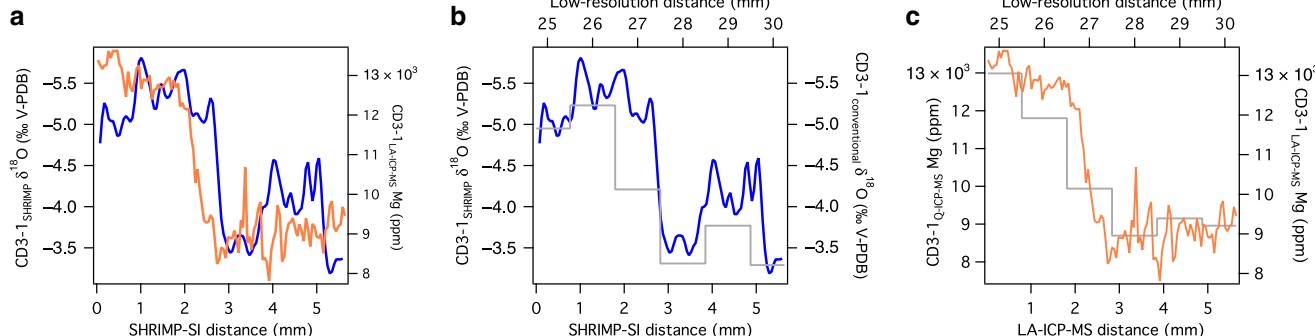

**Fig. 4 High-resolution data from CD3-1 for Termination II. a** SHRIMP-SI ion-microprobe $\delta^{18}O$ (blue) and laser-ablation (LA) ICP-MS Mg (orange) series plotted on a common distance scale. The LA-ICP-MS measurements were made on the exact spot positions of the SHRIMP-SI analyses (see Methods). SHRIMP-SI data are shown as a three-point Gaussian smooth of the raw series (see Supplementary Fig. 4). **b** Comparison between the high-resolution SHRIMP-SI $\delta^{18}O$ (blue; measured by ion-microprobe) and the low-resolution $\delta^{18}O$ data (grey cityscape line plot; measured conventionally). **c** Comparison between the high-resolution LA-ICP-MS Mg (orange) and low-resolution Mg (grey cityscape line plot; measured by solution ICP-MS).

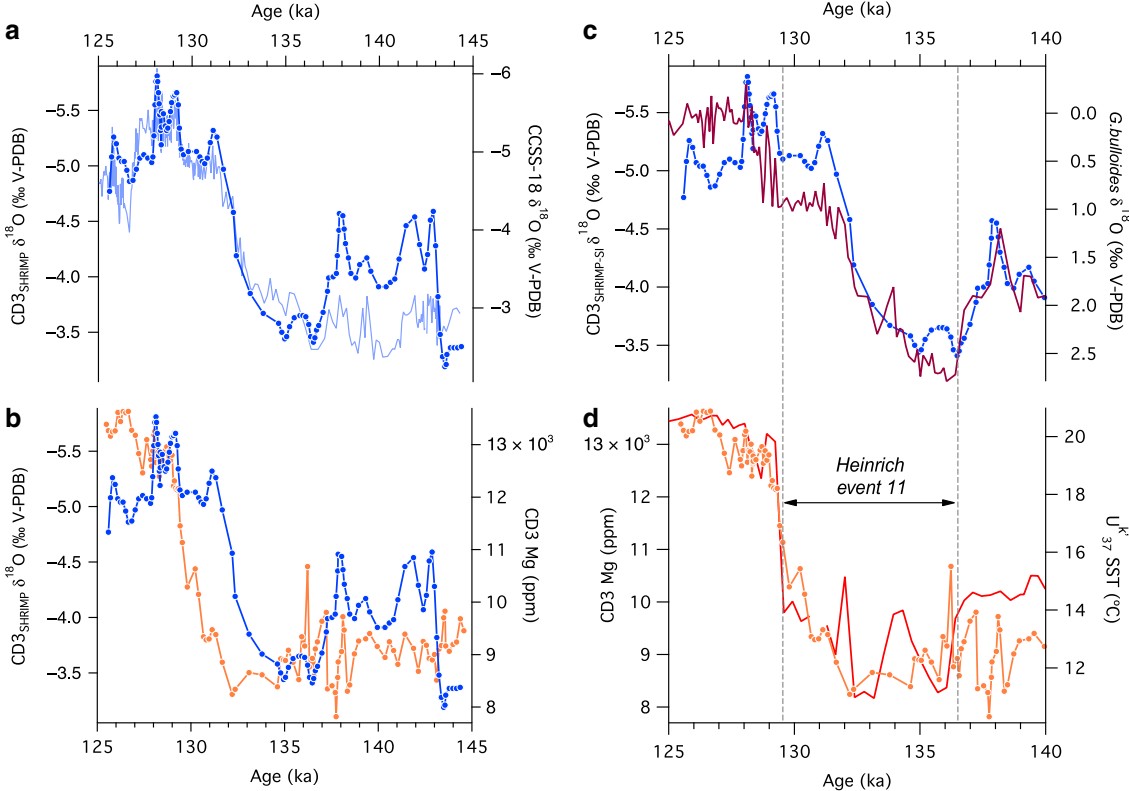

**Fig. 5 CD3 and ocean-core data for Termination II. a** SHRIMP-SI $\delta^{18}O$ (dark blue with symbols) synchronized to the Corchia Cave speleothem $\delta^{18}O$ stack[40] (CCSS-18: light blue) enabling the U-Th chronology from CCSS-18 to be transferred to the CD3-1 series. See 'Methods' and Supplementary Fig. 6. **b** SHRIMP-SI $\delta^{18}O$ (blue) and laser-ablation (LA) ICP-MS Mg (orange) series plotted on the CCSS-18 chronology, highlighting the phasing of the large Mg increase and $\delta^{18}O$ decrease across Termination II. **c** Comparison between the planktic $\delta^{18}O$ (G. bulloides) (brown) from the Iberian margin and the high-resolution SHRIMP-SI $\delta^{18}O$ from CD3-1 (blue with symbols). The ocean record is shown on the chronology of Tzedakis et al.[40]. **d** Comparison between the high-resolution LA-ICP-MS Mg CD3-1 data (orange with symbols) and Iberian-margin SST (red). The correlation between the Mg and SST series is statistically significant ($r = 0.92$; $p < 0.05$; df = 36; see Supplementary Fig. 7). The grey panel marks the position of Heinrich Event 11.

stalagmite $\delta^{18}O$ (Fig. 5a). Anchoring of the CD3-1 data to the CCSS-18 age model[40] (see 'Methods' and Supplementary Fig. 6) shows that the phase difference between the $\delta^{18}O$ and Mg evident in Fig. 4a equates to several thousand years (Fig. 5b).

A previous study from Corchia Cave suggested that speleothem $\delta^{18}O$ across T-II tracked the rise in Iberian-margin SSTs[49]. This interpretation was subsequently revised by Marino et al[53]., who argued that the speleothem $\delta^{18}O$-SST link was compromised during the T-II meltwater event, Heinrich 11 (H11). Here, meltwaters from the collapse of the Northern Hemisphere ice sheets reached the Iberian margin and western Mediterranean, causing a decrease in surface ocean $\delta^{18}O$. This was captured by air masses reaching Corchia Cave, leading to decreased speleothem $\delta^{18}O$ at a time of cooling, not warming. As Fig. 4a shows, the phasing of CD3-1 Mg and $\delta^{18}O$ provides first-order support for a strong meltwater effect on the speleothem $\delta^{18}O$ during H11, which precedes the temperature rise through the remainder of the termination. We now test this further by comparing the CD3-1 data to ocean-sediment proxies from the Iberian margin.

The MD01-2443/2444 ocean record (Fig. 1) through T-II has already been synchronized to the CCSS-18 chronology based on aligning the speleothem $\delta^{18}O$ with the temperate tree-pollen abundance series from the ocean sediments[40]. The CD3-1 $\delta^{18}O$ data are in excellent structural agreement with the planktic $\delta^{18}O$ from G. bulloides for the period up to the end of H11 (Fig. 5c). If the CD3-1 Mg was faithfully recording regional temperature changes, it should align with SSTs from the same ocean record.

As Fig. 5d shows, the correspondence between the Mg and SSTs is strong (Pearson's $r = 0.92$, $p < 0.05$, df:= 36; see also Supplementary Fig. 7), particularly across the sharp rise at the end of H11. This provides firm evidence that CD3-1 preserves a record of regional temperature changes through T-II.

In summary, high-resolution Mg and $\delta^{18}O$ series from CD3-1 show similar patterns through T-II, but significant decoupling occurs during the crucial interval of H11. The influence of SST on rainfall amount and, ultimately, speleothem $\delta^{18}O$ at the cave site is overprinted by the moisture-source effect induced by the large meltwater pulse from ice-sheet collapse. In addition to the earlier points made about speleothem $\delta^{18}O$ during cold sapropels, this suggests that the subaqueous speleothem Mg is a far more consistent, robust and more direct palaeotemperature proxy than $\delta^{18}O$, particularly during millennial-scale Heinrich events.

## Discussion

The tendency for CD3-1 Mg to show patterns consistent with forcing by source-water temperatures is unprecedented for a speleothem. Although early studies proposed that Mg palaeo-thermometry could be applied to speleothems[73,74], subsequent research has convincingly demonstrated otherwise[19,75], with many examples linking Mg variations to hydrological changes[20–28]. However, these studies have only been conducted on stalagmites and flowstones[18], where the calcite is deposited relatively rapidly from a thin film of percolation water that can degas $CO_2$ very efficiently. This suggests the subaqueous depositional environment

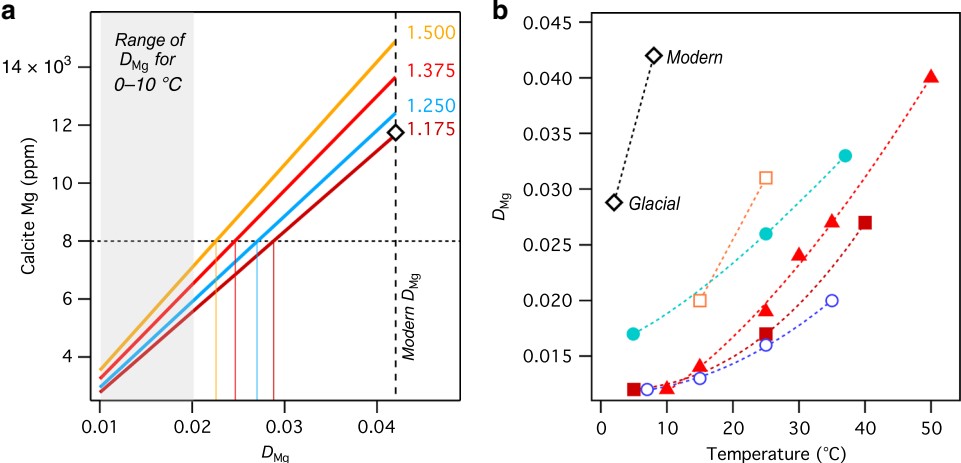

**Fig. 6 Modelled magnesium partition coefficients from CD3. a** CD3 Mg concentrations derived from a range of Mg partition coefficients ($D_{Mg}$) and four hypothetic pool-water Mg/Ca ratios (values in mol/mol displayed next to each line). The modern $D_{Mg}$ (0.042) is shown by the long-dash vertical line. The black diamond shows the mean modern calcite concentration value (~11,750 p.p.m.) and the mean pool-water Mg/Ca value (1.175 mol/mol)[33]. The short-dash horizontal line is the minimum measured Mg concentration (8000 p.p.m.) from Fig. 5d. Each coloured solid vertical line gives the $D_{Mg}$ value for the corresponding pool-water Mg/Ca at 8000 p.p.m. The grey panel shows the range of $D_{Mg}$ values for 0–10 °C from previously published studies[33]. **b** Published experimentally derived $D_{Mg}$–temperature relationships (coloured symbols) compared to CD3 (black diamonds). Open and closed coloured symbols are from cave-analogue and seawater-analogue experiments respectively (green solid circles, Burton and Walter[77]; red solid triangles, Oomori et al.[78]; solid brown squares, Mucci[79]; open blue circles, Day and Henderson[80]; open orange diamonds, Huang and Fairchild[81]). The modern CD3 value is based on Drysdale et al.[33]; the glacial CD3 value is the maximum possible $D_{Mg}$ assuming a MIS 6 glacial cave temperature of 6 °C below present (i.e., 2 °C) and a pool-water Mg/Ca value (1.175 mol/mol) equivalent to modern.

of CD3 is of vital importance in enabling pool-temperature changes to drive speleothem Mg variations.

Percolation waters reaching the Galleria delle Stalattiti are of low ionic strength and relatively invariant hydrochemistry[33,42,76], characteristics attributed to the thin soils above the cave and the great thickness of the overlying host rock. A portion of these percolation waters drains to Laghetto Basso pool. The thicker water column of the pool (maximum depth: 1.3 m)[33] would promote slower $CO_2$ degassing compared to the thin water films from which stalagmites and flowstones grow[14], where the rate of $CO_2$ removal would be greater. The pool waters are also of low calcite saturation state[33]. Taken together, this would explain the very low growth rates in CD3 (mean: ~0.3 mm;kyr$^{-1}$)[33], which are lower by order of magnitude or more compared to those of nearby stalagmites[30,43,47,49,58].

The predominance of thermal, rather than hydrological, forcing of CD3-1 Mg over G–IG time scales suggests either that the Mg/Ca of the water (Mg/Ca$_{[aq]}$) reaching the cave chamber does not change significantly over such time scales and/or that the slope of the temperature dependence of the Mg partition coefficient is significantly greater for a subaqueous speleothem compared to subaerial speleothems or other calcites[33]. Although the hydrochemistry of modern percolation waters reaching the chamber are stable at the decadal scale[33,42], some degree of pool Mg/Ca$_{[aq]}$ variability would be expected to occur over a full glacial termination, given the likelihood of atmospheric reorganisation and associated hydrological changes. Mg data from Corchia stalagmite CC5, collected within metres of CD3-1, and which also grew through T-II, reveal decreasing values through T-II[49] (Supplementary Fig. 8). This change is broadly the opposite to what is seen in CD3-1 over the same interval. This suggests forcing by hydrologically driven variations in drip Mg/Ca$_{[aq]}$; the negative covariation between CC5 Mg and U (another palaeo-hydrological proxy[49]) supports the argument for moisture-driven changes in drip Mg/Ca$_{[aq]}$ (Supplementary Fig. 8). Such changes would also be registered in Laghetto Basso waters, the local sink of drip waters in the Galleria delle Stalattiti. The fact that Mg

partitions into the pool and stalagmite calcites differently, in spite of being derived from essentially common source waters, strongly suggests the exceptionally slow growth rate of subaqueous CD3 is responsible for a stronger temperature dependence of the Mg partition coefficient ($D_{Mg}$).

The modern $D_{Mg}$ in CD3 (0.042 ± 0.002) is about three times larger than coefficients derived from other natural (but non-subaqueous) speleothem types (or laboratory analogues thereof) precipitated at similar temperatures[33]. Stronger temperature dependence compared to other calcites could override the effects of changes in pool Mg/Ca$_{[aq]}$. To test this, we modelled a range of $D_{Mg}$ and hypothetical pool Mg/Ca$_{[aq]}$ values to determine the amount of change in each that is necessary to generate the CD3-1 Mg values observed during glacial maxima. The lowest Mg values observed over the last 350 kyr approach 8000 p.p.m. just prior to and during H11 (Fig. 5b). The modelling indicates that a $D_{Mg}$ value of 0.029 is required to reach 8000 p.p.m. of calcite Mg based on the modern Mg/Ca$_{[aq]}$ of 1.175 mols mol$^{-1}$ (Fig. 6a). The evidence from stalagmite CC5 of hydrologically driven changes in Mg/Ca$_{[aq]}$ (Supplementary Fig. 8) suggests it is highly unlikely that pool-water Mg/Ca$_{[aq]}$ would be equivalent to or lower than modern values due to glacial climates at Corchia being drier than interglacials[43,49]. We can therefore assume that a $D_{Mg}$ of 0.029 is the maximum possible value. The more plausible scenario of higher glacial pool-water Mg/Ca values would reduce $D_{Mg}$ values in CD3 to less than 0.029. For example, a hypothetical pool-water Mg/Ca of 1.375 and a glacial pool-water temperature 6 °C below present would give a $D_{Mg}$ of 0.025 (red curve, Fig. 6a). Comparing the glacial and modern $D_{Mg}$ values and pool temperatures confirms that the slope of the temperature-$D_{Mg}$ relationship in CD3 is steeper than those derived from previous studies[77–81] (Fig. 6b). This suggests that the strong temperature dependence of Mg partitioning from the pool water to CD3 is the principal reason why the Mg in this speleothem tracks temperature, rather than hydrological, change.

Having established why CD3 Mg tracks pool-water temperature rather than hydrology, we now explore what exactly the

pool-water temperature represents in terms of the external temperatures. The air temperature of a deep-cave environment is usually stable (to within a few tenths of a degree centigrade) and, in most studies, is assumed to be representative of external MAAT[14]. However, caves often extend over large altitude ranges and contain multiple entrances, so the notion of MAAT is potentially ambiguous, and has rarely received critical analysis[82]. The long residence time of the Laghetto Basso pool waters (at least 1 year)[33] means that near-thermal equilibrium is reached with the gallery air temperature[42]. In the deep galleries of vadose cave systems such as Corchia Cave, where natural entrances are numerous but small, air temperatures are modulated by the weighted mean temperature of infiltration waters[82] which will deviate from the MAAT according to the seasonality of local rainfall. Rainfall data from the meteorological station of Retignano (440 m.a.s.l.), located a few kilometres from Corchia Cave, show an amount-weighted mean annual temperature of precipitation of 12.5 °C. This is 1.5 °C lower than the MAAT measured at the station and reflects the predominance of a cool-season rainfall regime characterizing this part of the Mediterranean region. Taking the same 1.5 °C offset, the Laghetto Basso pool-water temperature of ~8 °C suggests that the MAAT at the average recharge altitude is 9.5 °C. Using the local lapse rate of 4.8 °C 1000 m$^{-1}$ (Supplementary Figure 2) gives a mean recharge altitude of ~1450 m.a.s.l., in agreement with the mean estimate (1500 m.a.s.l.) based on the local $\delta^{18}$O-altitude gradient[45]. Thus, we conclude that variations in pool-water temperature will reflect the external amount-weighted temperature of precipitation in the vicinity of ~1450–1500 m.a.s.l.

Future high-resolution Mg analysis of CD3-1 over its entire length could yield a quantitative record of regional temperatures comparable to those produced from ocean sediments and ice cores. This would require a Mg-temperature transfer function. The best prospect for such a function would be the clumped-isotope palaeothermometer[83]. Clumped-isotope palaeotemperatures recently determined from the actively growing surface of CD3 agree with modern pool temperatures and constrain a two-point temperature-calibration equation anchored at 34 °C by a similar slow-growing subaqueous speleothem at Devils Hole, USA[44]. Applying a Mg-clumped-isotope temperature calibration function that incorporates the full range of Pleistocene climate states to laser-ablation inductively coupled plasma mass spectrometry (LA-ICP-MS) Mg measurements could yield a quantitative high-resolution temperature series from CD3. This series would be produced far more efficiently than one based on clumped-isotope palaeothermometry alone. However, this would be conditional upon either ensuring that hydrologically driven changes in Mg/Ca have an insignificant impact on CD3 Mg, or developing the means to correct for such changes: this is the subject of ongoing research. Nevertheless, a semi-quantitative Mg-based temperature reconstruction is already possible, paving the way for nearby ocean records to be tied to a radiometric chronology via Mg-SST synchronization. There are significant implications for developing such ocean records. For instance, it would enable G–IG and millennial climate changes recorded in ocean sediments to be placed on an independent time scale, free of assumptions regarding astronomical forcing. This would allow for the robust interrogation of the mechanisms that force these palaeoclimate changes.

Finally, in terms of the broader implications of this work, our results suggest subaqueous speleothems offer a previously unrecognized opportunity for generating continental palaeo-temperature records beyond the polar regions. Standing-water bodies are widespread features of cave systems, and many contain subaqueous speleothems. However, these have been consistently overlooked as palaeoclimate archives in favour of stalagmites and flowstones. Given that the effects of climate change on pool chemistry should vary in sync with those of drip waters feeding the pool (i.e., according to local geology, host-rock thickness, and aquifer architecture)[14], we suggest two key site-selection criteria based on our Laghetto Basso example[33]. First, the pool water needs to be of low-ionic strength and low saturation: this combination guarantees slow calcite growth rates. The changes in Mg through T-II reported from stalagmite CC5 and subaqueous speleothem CD3-1 are antiphase, with clear first-order hydrological and temperature signals recorded respectively, in spite of similar source-water chemistry. The order of magnitude (or greater) higher growth rates of CC5 compared to CD3-1 suggest the stronger temperature dependency of the Mg partition coefficient in CD3-1 is attributable to the kinetics of slow calcite precipitation[33]. Second, the Mg/Ca of the source waters should vary within a relatively narrow range to prevent the temperature signal being compromised. For example, temperatures could be overestimated during warm-dry periods (when drying would enhance PCP or IDD, and raise Mg/Ca$_{[aq]}$, thus exacerbating the increased partitioning of Mg due to warming) and underestimated during warm-wet periods (where increased recharge would minimize PCP or IDD, and supress Mg/Ca$_{[aq]}$, dampening the increase of Mg due to warming). Given that deep glacial and optimum interglacial climates are often associated with extremes of low and high rainfall, respectively, any hydrological influence on partitioning would dampen the amplitude of Mg-based G–IG temperatures. Low-ionic-strength cave waters are less susceptible to PCP or IDD because low Ca$^{2+}$ concentrations would lead to lower calcite precipitation rates and thus less removal of Ca$^{2+}$ en route to the pool system[84], causing minimal change in Mg/Ca$_{[aq]}$ at the pool. Furthermore, the deeper a subaqueous speleothem sits within a cave system, the greater the flow path along which the percolation waters must traverse, and the closer they evolve to the point where little additional detectable calcite precipitation can occur under the full range of hydrological conditions. This would also result in minimal changes in Mg/Ca$_{[aq]}$ in the pool, thus supressing the hydrological influence on Mg partitioning in the subaqueous speleothem. Finally, as has been shown previously[31,33] as well as in this study, the issue of Th scavenging and its effect on U-Th age accuracy is likely to confound attempts to produce independent chronologies in subaqueous speleothems. However, as we have shown in this study with the high-resolution T-II data, it is possible to overcome this issue by synchronizing stable isotope data from coeval stalagmite and subaqueous speleothem records and applying the stalagmite U-Th chronology.

## Methods

**Low-resolution stable isotope and Mg data from CD3-1.** Low-resolution stable isotope ($\delta^{13}$C and $\delta^{18}$O) data for CD3-1 were produced by sampling the upper 71 mm of the core continuously at 1 mm intervals using a Taig CNC micromilling machine (School of Geography, University of Melbourne) fitted with a 1 mm end-mill bit. The $\delta^{13}$C and $\delta^{18}$O were measured on 0.7 ± 0.1 mg aliquots of these powders using a GV Instruments GV2003 continuous-flow isotope ratio mass spectrometer housed at the Discipline of Earth Sciences, The University of Newcastle, Australia[49]. The residual powders were prepared for trace element measurement at the Australian Nuclear Science and Technology Organisation (Sydney, Australia) using a Varian 820MS quadrupole inductively coupled plasma mass spectrometer. A series of standards similar in concentration to the samples were used to calibrate the instrument. All standards were prepared from certified 1000 ± 3 p.p.m. single-element National Institute of Standards and Technology (NIST) standards in 3% v/v Merck Suprapure HNO$_3$ spiked with Li, Sc, Rh, In, Re, Bi. The spikes are used as an internal standard correction for instrument drift, matrix correction and plasma fluctuations on the ICP-MS. A certified cocktail, also NIST traceable, from another supplier was used as an independent quality-control check at the beginning of the run. A set of standards was also analyzed after every 20th sample to monitor instrument drift. All carbonate powders were dissolved in the same 3% v/v nitric acid solution above in the ratio of 0.1 mg/mL to minimize matrix and concentration variation. All the masses of the sample and solution used were recorded and calculated for the dilution-factor correction, which was applied to the solution concentration to give a final concentration of the calcite samples. All

ICP-MS data were normalized to $^{43}Ca$. High-purity water (18.2 MΩ cm) was used for the preparation of all solutions, and a separate solution containing 0.25 mL of 1% v/v Triton X-100 in 5 L of 1% $HNO_3$ was used for rinsing between standards and samples. Uncertainty on the Ca and Mg measurements is ±1.6%. Results are reported in Supplementary Data 2.

**U-Th dating and age modeling**. The low-resolution $δ^{13}C$, $δ^{18}O$ and Mg data were anchored to a radiometric chronology based on 121 U-Th age determinations (Supplementary Table S1), which were made using multi-collector inductively coupled plasma mass spectrometry[85,86]. Uncorrected ages were calculated using the latest U and Th decay constants[87], then corrected based on an estimate of the initial $^{230}Th/^{232}Th$ activity ratio $((^{230}Th/^{232}Th)_i)$ of 14 ± 9 derived from measurements made on water samples collected from Laghetto Basso in 2015 (shown in blue in Supplementary Table S1). Twenty-two ages are from a previous study[30] (shown in black in Supplementary Table 1) and were extracted from CD3-1 at consecutive intervals of ~0.5 mm using the micromilling technique described in the same paper. Six identically extracted samples not part of the earlier study[30] are reported here for the first time and extend the original sequence of ages further back in time (CD3-200 to CD3-205: shown in grey in Supplementary Data 1). The remaining 93 samples were extracted using a 1 mm end-mill bit fitted to a TAIG CNC micromilling lathe (School of Geography, University of Melbourne). With the exception of samples CD3-2000 to CD3-2034, these were drilled or milled from core CD3-1 (shown in green in Supplementary Table S1), so that their positions could be registered directly to the stable isotope and Mg series. The other 35 samples (shown in orange in Supplementary Table S1) were drilled in two traverses from a second core (CD3-2), with the break occurring between CD3-2014 and CD3-2015. Due to a drilling position error on the lathe, CD3-2015 (the first sample of the second traverse) was centred 1.25 mm, not 1.00 mm, after the previous sample. The thickness of the section in core CD3-2 was determined to be 28.7% larger than the corresponding section in CD3-1. Accordingly, we carried out a depth adjustment by dividing the original CD3-2 depth centres by 1.287 to bring these dating positions onto the same scale as CD3-1.

From the total of 121 U-Th ages, 7 were identified as outliers (highlighted in grey shading and italicized text in Supplementary Table 1 and red symbols and error bars in Supplementary Figure 1) and have been removed from the depth-age model. An age model was subsequently produced from 114 U-Th ages using the finite-positive-growth-rate-method[47,88], with the corresponding age uncertainty versus age plot for the period 0–350 ka (see Supplementary Fig. 1).

**Ocean-sediment data for the low-resolution study**. We first compared the low-resolution CD3-1 stable isotope and Mg series to the 'LR04 benthic stack'[51] (Fig. 2) on its originally published age model. We then compared the CD3-1 series to the ocean-sediment data from Iberian margin cores drilled from sites MD01-2443/2444[35–40] (Figs. 1 and 3) using an age model based on a synchronization to the synthetic Greenland ice-core[89], except for the period 140–110 ka, which is based on a synchronization of the temperate tree pollen series from the ocean record to the Corchia Cave speleothem $δ^{18}O$ stack (CCSS-18)[40]. To optimize the comparison between the low-resolution CD3-1 data and the much higher-resolution Iberian margin record, we resampled and synchronized the two records using the following steps. First, we resampled each ocean time series at a 200-year resolution, then binned the data by averaging consecutive time slices of 25 data points to produce time series at 5 kyr increments. This bin size approximates the average sampling interval of the low-resolution CD3-1 data. Second, we synchronized the CD3-1 $δ^{18}O$ series to the Iberian margin record using the $δ^{18}O$ of the planktic foraminifer *G. bulloides* as the tuning target. The choice of planktic $δ^{18}O$ as the tuning target is explained in the main text. The purpose of the synchronization was to overcome the inaccuracy inherent in the CD3-1 U-Th age model (see main text). The synchronization was implemented in the software *Analyseries*;[90] the result and tuning points are shown in Supplementary Fig. 3. Third, to enable point-by-point correlations between the CD3 and ocean data (shown in Fig. 3c, d), we then resampled the tuned CD3-1 series at the same 5 kyr interval. The data are reported in Supplementary Data 2.

**High-resolution SHRIMP-SI $δ^{18}O$ analysis of CD3-1**. High-resolution $δ^{18}O$ analyses were conducted on the T-II section of CD3-1 using the Sensitive High-Resolution Ion Microprobe–Stable Isotope (SHRIMP-SI)[91] housed at the Research School of Earth Sciences, The Australian National University. The target section was cut from the core, placed in a 25 mm-diameter circular mould and set in epoxy resin. Included in the resin disc were aliquots of two stable isotope standards: the international standard reference material NBS19 ($δ^{18}O$ −2.20‰ V-PDB) and a University of Melbourne stable isotope laboratory calcite standard (MELB1: $δ^{18}O$ −4.99‰ V-PDB) previously calibrated against NBS19, and another international standard reference material, NBS18. The disc was trimmed to a thickness of 4 mm, and the surface polished then gold coated prior to analysis.

The $δ^{18}O$ traverse comprised a series of 30 μm spots, with centres spaced ~50 μm apart. The spot positions were identified using on-screen digitizing software; imperfections in the speleothem surface, such as fractures, were avoided by moving the spot position laterally. The analytical conditions on the SHRIMP-SI were similar to those reported by Ickert et al[91]. A 15 keV $Cs^+$ primary beam was used to sputter the sample. Sample charging was neutralized by focusing a 1.9 keV electron beam

onto the sputter site. O-secondary ions were extracted to real ground (ca. 10 keV resultant energy) and the secondary beam was focused and steered through a quadrupole triplet system to the source slit of the SHRIMP-SI mass analyser.

Oxygen isotopes were analysed in multiple collection mode with a 400 μm slit for $^{16}O^-$ (mass resolution ca. 2000 M/ΔM at 10% peak height) and a 300 μm slit for $^{18}O^-$ (ca. 3000 M/$ΔM_{10\%}$), which was sufficient to resolve interferences such as $^{16}OD$ and $^{17}OH^-$. Ion beam intensities were measured by Faraday cup and electrometer in current mode, with feedback resistors of $10^{11}$ Ω and $10^{12}$ Ω for $^{16}O^-$ and $^{18}O^-$ channels, respectively. Detector backgrounds were measured prior to each spot. Each analysis consisted of two sets of 6 scans, with each scan comprising 10 two-second integrations. Electron-induced secondary-ion emission count rates[91] were measured three times during the analysis: at the start of each set and at the end of the analysis. Analyses of the standards were interspersed with analyses of the speleothem.

The final sample $δ^{18}O$ data ($n = 117$, after removing overlapped sections) were normalized to the V-PDB scale using NBS19, then scale-corrected for mass fractionation using a two-point linear adjustment between the NBS19 and MELB1 values. External reproducibility on NBS19 and MELB1 were 0.5‰ and 0.4‰, respectively (1σ). The raw data, along with a three-point Gaussian smoothing spline, are plotted in Supplementary Fig. 5 and reported in Supplementary Data 2.

**High-resolution LA-ICP-MS analysis of CD3-1**. Following the SHRIMP-SI analyses, the gold-coating was removed from the resin disc by polishing with 0.5 μm aluminium oxide powder, and the sample prepared for trace-element measurements (Mg, Sr, Ba, U and Zn). Analysis was implemented directly on the SHRIMP-SI spots using LA-ICP-MS at the School of Earth Sciences, The University of Melbourne[92]. Only the Mg data are reported here. Prior to analysis, the disc was twice ultrasonically cleaned in de-ionized water for 15 min, before being placed into the laser chamber. The SHRIMP-SI spots were readily identified using a video camera, and their positions digitized using *Geostar* software[92]. Each spot was ablated at a rate of 5 Hz for 50 s using a 26 μm circular spot. Trace-element abundances in counts per second were converted to parts per million by normalization to the NIST610 international reference standard using *Iolite* software[93]. Final mean (±2 SE) trace element concentrations were calculated from an 18 s time window that commenced two seconds into each ablation (to remove surface contaminants) and ended at $t = 20$ s due to downhole loss of sensitivity and mass fractionation. Although the NIST610 reference material has Mg concentrations much lower than CD3 means (Mg = 432 ± 9 p.p.m., based on the 2011 GeoREM values: http://georem.mpch-mainz.gwdg.de), previous work has shown that the concentration values are within 10% of those measured by matrix-matched ICP-MS solution analysis from the outer surface of CD3[33]. The raw data, along with a 3-point Gaussian smoothing spline, are plotted in Supplementary Figure 5 and reported in Supplementary Data 2.

**CD3-1 chronology through T-II**. Given the issues with the CD3-1 U-Th chronology, the SHRIMP-SI $δ^{18}O$ was synchronized to the U-Th time scale of the CCSS-18 $δ^{18}O$ record[40] (Fig. 5a). This was derived from stalagmites collected from the same chamber as CD3-1[47,49]. The synchronization was implemented using *Analyseries*;[90] the result and tie points are shown in Supplementary Fig. 5. Linear interpolation was used to assign a CCSS-18 age to each CD3-1 depth position. Since the SHRIMP-SI $δ^{18}O$ and LA-ICP-MS Mg have identical depth positions, the Mg series is automatically registered to the same age model (Fig. 5b).

**Ocean data from Site MD01-2444 through T-II**. We compared the CD3-1 $δ^{18}O$ and Mg data through T-II with the planktic $δ^{18}O$ and $U^{k'}_{37}$ alkenone SST record from site MD01-2444[3,40] (Fig. 1). This ocean record had already been tied to a radiometric time scale by synchronizing the ocean-core's temperate tree-pollen abundance series to the CCSS-18 $δ^{18}O$[40]. This enables direct analysis of whether CD3-1 Mg is capturing regional SST changes. To determine the correlation between Mg and SST, we first resampled each of the CD3-1 and ocean-core series at an increment of 100 yr then binned the data by averaging consecutive time slices of five data points to produce time series at 500-yr increments, approximating the largest individual age step in the SST series. The correlations between the CD3-1 and ocean data are shown in Supplementary Fig. 8 and the data are reported in Supplementary Data 2.

## Data availability
All data produced in this paper are available in the Supplementary Data files.

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

## Acknowledgements

We are grateful to the Gruppo Speleologico Lucchese for their assistance with the recovery of the CD3-1 drill core. This study was funded by Australian Research Council Discovery Project grants DP0773700 (to R.D., J.H. and G.Z.), DP110102185 (to R.D., J.W., J.H. and G.Z.) and DP160102969 (to R.D., J.W., G.Z., E.R. and P.T.). I.C. was the beneficiary of a research associate position funded under Australian Research Council Discovery Project DP0773700.

## Author contributions

R.D. designed the study in conjunction with J.H., I.C., G.Z., I.I. and E.R. R.D., I.C., G.Z., L.P. and I.I. were involved in the collection, cutting and/or sampling of the CD3-1 core. R.D. and I.C. ran the low-resolution conventional stable-isotope analysis. H.W. ran the low-resolution solution ICP-MS analyses. T.I., P.H. and R.D. ran the high-resolution SHRIMP-SI analyses following an initial concept study by J.H. T.I. and P.H. processed the data. R.D., A. Greig and J.W. ran the LA-ICP-MS Mg analyses and processed the data. R.D. and I.C. carried out the U-Th sampling, J.H. and I.C. performed the U-Th chemical preparation, and J.H. ran the U-Th age measurements and reduced the data. J.H. and E.C. performed the age modelling. A. Govin helped R.D. with the synchronizations using *Analyseries*. R.D., I.C., A. Govin and P.T. discussed the results. R.D. wrote the paper with input from all co-authors.

## Competing interests

The authors declare no competing interests.
