## [Peer Review File · Nature Communications]

Reviewers' comments:

Reviewer #1 (Remarks to the Author):

This is a very interesting study arguing that Mg concentrations in a subaqueous calcite deposit within a well-studied cave site in Italy represent temperature within the pool at the time of deposition. Mg has long been linked with water temperature in marine carbonates, and early promise within stalagmites did not develop into anything particularly significant because drip water Mg variability (for sub-aerially-deposited stalagmites) was much more pronounced than any temperature dependency. Here, the calcite was deposited in a cave pool in a deep cave where the water temperature is apparently coupled to cave chamber temperature, which in turn reflects mean annual outside temperature. The authors argue that essentially no other control is relevant in this situation, and that therefore water temperature is directly linked to outside air temperature, and that this can be reconstructed using the Mg in the calcite.

There are a lot of positives here. I agree with the authors that the simplest explanation for the Mg-temperature relationship is that Mg incorporation into the calcite is probably controlled by pool temperature. This would be an important result because, as the authors state, temperature reconstructions on land over G-IG timescales are rare, and this would provide a reasonably straightforward additional temperature proxy. Most published speleothem records are from sub-aerial stalagmites, but cave pools themselves are not rare (I think that the authors should note this in the text) and may represent a very underutilised resource, with the potential to reconstruct temperature across a wide variety of environments. The correlation with temperature across T-II is impressive.

However, I feel that there needs to be some more discussion included before the manuscript could be publishable in Nature Communications. My major comments revolve around the fact that I do not feel that as currently written the manuscript makes a definitive case that temperature controls Mg incorporation into the pool calcite. As I mentioned above, I do agree that this is probably what's happening, but there needs to be more evidence, and more discussion of temperature in the main text. As the authors state on lines 118-119, this may be the first time that Mg in a speleothem has been shown to vary positively with temperature, so this is a potentially controversial result; it therefore needs to be supported carefully. For example, it would be great to emphasise the temperature inside the cave pool and chamber in the main text. There is surprisingly little discussion about the relationship between pool water temperature and outside MAT; I appreciate that there is more discussion in other publications, and some in the methods, but this should be highlighted in the main text. How many temperature measurements were made of the water and of the cave air? Drip water temperature measurements? How stable is the pCO₂ of the cave chamber? Are there no measurements of temperature directly outside the cave entrance? How far away is the Levigliani met station? What is the seasonality like in all these parameters, if present? Could the cave chamber PCO₂ be different during the last glacial than at present? What effect might this have on pool Mg concentrations, since PCO₂ would affect Prior Calcite Precipitation amounts?

Another point that should be discussed further is the excess ²³⁰Th issue, and whether it will be relevant to other similarly deposited calcites. The authors very correctly identify this as an issue, and it has been previously identified as an issue at Devil's Hole (also noted by the authors), another subaqueous speleothem. Is this likely to be a universal issue with subaqueous speleothems? I don't see any reason why it wouldn't be, and if this is the case, it may not be a simple task to use these sorts of speleothem to correct sea sediment and glacial ice core derived temperature reconstructions via wiggle matching. The chronology here was corrected using wiggle matching to sea sediment records; if the excess Th issue is universal to subaqueous speleothem, can the data be corrected without resorting to synchronisation?

Line 47: ‘...on G-IG timescales are sparse...’ – tree rings of course are excellent palaeotemperature proxies but on a shorter timescale, so worth quantifying the timescales you are discussing here.
Line 79: other references here: Moseley et al., Science (2016), and Wendt et al., Science Advances (2018).

83: ‘concomitant’

120: no need for quotation marks here

130-133: It’s good that the authors have taken this into account.

The high-resolution data over T-II are a nice inclusion, but with a 0.5 per mil uncertainty in the $\delta^{18}\text{O}$ values, are the SHRIMP values interpretable, other than the large shift at 2.5 mm?

234: The cave temperature is not always equal to the outside MAT. Different caves can behave differently, and cold traps can exist when colder (denser) winter air fills a cave, causing cave temperature to be somewhat lower than outside MAT. Again, are any cave air temperature data available for the site? If so, could they be included here? The constant cave air pCO_2 (could the authors list the value) suggests that cave air temperature is similarly constant, as suggested by the authors, but this should be discussed as it is the crux of the manuscript.

256: Could the growth rates be listed here?

323: “has the potential to record external air temperature changes above the cave.” Again, as mentioned above - are conditions at the cave site comparable to outside conditions? Methods discuss the lapse rate and calculates an outside temperature at the site. This makes sense, but are no direct measurements available? These would obviously be preferable to a calculation. The key point of the manuscript is direct temperature reconstruction using Mg; the key information regarding the site temperature should not be relegated to the Methods – this should be in the main text, and it’d be nice to see more information regarding the local temperature signal, like seasonality.

Reviewer #2 (Remarks to the Author):

This paper uses a series of comparisons between Mg content in a subaqueous speleothem and geochemical proxies from the speleothem and an ocean sediment core to argue that the speleothem Mg concentration is a function of mean annual air temperature. The major claims of this paper are:

- a. That the CD3-1 speleothem Mg concentration accurately records pool-water temperature rather than hydrological variability.
- b. That the Mg record can be used to synchronise ocean core records to radiometric timescales.
- c. That the partition coefficient for CD3-1 Mg/Ca from pool water to calcite is much higher than is normal for speleothems and that the slope of the relationship between the partition coefficient and temperature is much steeper.

As far as I am aware, these claims are novel. The paper implies that claims a and c may be generalised to other subaqueous speleothems, but as yet there have been no studies to verify this. The paper certainly has major implications in this regard, though significant further work will be required to test these claims in other subaqueous speleothems. If it is found that subaqueous speleothems in general accurately record cave temperatures (and therefore mean annual temperatures outside the cave), this will be an exciting step forward for speleothem palaeoclimate, as it will allow well-dated reconstructions of mean annual air temperature in terrestrial regions. As

stated in the article, at present such reconstructions are difficult to produce from speleothems as the geochemical signals in speleothem calcite are affected by a range of different climatic and cave parameters.

Overall, I find the claim that CD3-1 Mg/Ca records temperature to be convincing given the evidence produced in the paper. However, I do have some comments, as detailed below.

1. A major comment I have with regard to this paper is its framing of the novelty of its conclusions. The Mg/Ca thermometer is initially validated with reference to d18O and d13C in the same speleothem, as these are interpreted as being related to temperature. Having stated that these temperature proxies already exist, the paper does not explicitly state why the Mg thermometer is of interest. I assume there is a closer relationship between the Uk37 palaeothermometer and Mg concentrations than there is between Uk37 and either d18O or d13C. However, I think that this should be explicitly stated and p-values/correlation coefficients should be produced for Iberian Margin Uk37 vs CD3-1 d18O and d13C (though these could be included in supplementary material). This would help to explain why the speleothem Mg thermometer is of special interest.

Towards the end of the paper (line 337), the authors also use the agreement between temperatures reconstructed from clumped isotopes and those reconstructed from Mg/Ca as validation for their argument that Mg/Ca records temperature. Once again, this raises the question of why use Mg/Ca at all if clumped isotopes tell us the same thing. Again, an explicit statement of the utility of Mg/Ca would be useful here.

Furthermore, a major plank of the "Implications" section is that Mg correlations with deep-sea palaeothermometers will allow ocean sediment cores to be placed on a radiometric dating scale. However, both this paper and earlier papers comparing Corchia Cave with ocean cores synchronise ocean sediments with radiometric time-scales by synchronising speleothem d18O with various ocean proxies, so it is not clear what advantage Mg gives in this regard. Again, an explicit statement is needed with regard to the utility of Mg concentrations over and above existing methods of correlation.

Finally, I think that a major implication of this paper is that subaqueous speleothems elsewhere might also have highly temperature-dependent Mg correlation coefficients and therefore might also record cave water temperatures (and therefore mean annual air temperatures outside the cave). These air temperatures are of great interest in and of themselves, not merely for the light they can throw on the timing of ocean sediment reconstructions – indeed, well-dated, high-resolution reconstructions of air temperatures in continental interiors would be very exciting. I feel that more could be made of this in the paper, both in the abstract and in the "Implications" section, which would clarify the novelty and the impact of the paper.

2. In Line 190, the authors state that the d18O and Mg proxies become decoupled at the termination of Heinrich 11 and that this is "critical", but they do not subsequently discuss the implications of the decoupling of these two proxies. Recharge rainfall d18O is previously identified in the paper as being related to temperature, so some kind of follow-up is needed here to suggest why these two temperature proxies become decoupled at this point and what the climatic reality of this might be. Again, the Mg proxy seems here to be giving us a much more detailed picture of the structure of climate variability at the end of Heinrich 11, so this has implications not just for our understanding of this climate event but for the utility of this proxy for reconstructing climate dynamics in general.

3. The argument about sapropels in lines 154-173 seems like it might have some important implications for the age model used in this section – are there other sapropels during the depositional period and might they have a similar problematic effect on the age model for CD3-1? As far as I understand it, the age model here for CD3-1 is based on the age model for the Iberian

Margin cores – how are the dates for S6 and S8 derived? Assuming there is some kind of external date associated with the sapropels, where should these sapropel events be located in terms of the G bulloides record from the IM cores, and do you actually see high d18O in the IM record at these times? If the low Mg and d18O in CD3-1 are assumed to be associated with S6 and S8, wouldn't it be better to use the known dates for these rather than, as the authors state, producing an alignment with incorrect features in the IM G bulloides record?

4. The paper is well-written but it is quite dense. In particular, the various records involved are synchronised to each other in several different ways to validate a series of arguments. Each of these synchronisations involves different levels and sources of uncertainty. I found this synchronisation and resynchronisation quite confusing and it was not clear to me what the implications of the various sources of uncertainty might be for the conclusions presented in the paper. For example, given the high temporal resolution of the reconstruction of the H11 termination, what are the implications of the 4 kyr uncertainty in the IM core age model? Given that one of the major thrusts of the paper concerns comparing records on common timescales, I think it is important to be clear about this aspect. I wonder if a bulleted list or table of synchronisations and their purposes and uncertainties could be included, perhaps in the supplementary material?

5. I am not a geochemist, but the question of the partition coefficient for CD3-1 seems to me to be very important. Is it generally known that subaqueous systems have higher/different partition coefficients than subaerial/thin film systems? If so, please state this and give references. If not, what might be the cause of this significant difference in partition coefficients and partition coefficient slopes? This is the key to why CD3-1 Mg responds to temperature rather than hydrology, and therefore the key to whether this may also happen in other subaqueous speleothems, so it has important implications for the potential impact of this study. I think it is worth a short discussion.

There follows a list of more minor comments.

Line 55 Mg/Ca in foram calcite also depends on species

Line 72 around → above?

Line 83 concomitants → concomitant

Line 130 the cause of the offset is most likely due to scavenging → the cause of the offset is most likely scavenging

Line 156-158 – black circle – do you mean black square?

Line 165 – intensive → intense

Line 168 – why do both types of sapropel produce reduced d18O?

Line 169-70 black circle or square? Do these two values represent both S6 and S8 or just S8?

Line 214 – I agree the structural agreement is good through the shaded portion, but it is significantly poorer from 125-129 ka. Probably not a big deal given that the argument hinges on H11, but worth being precise.

Line 218 Fig. S5 → Fig. S6

Line 237 – long residence time – quantify this

Line 238: ~thermal → “approximate thermal”

Line 241 patterns consistent with changes in → patterns assumed to be consistent with changes in

Line 250 is vital importance → is of vital importance for the response of calcite Mg to cave temperature [or similar]

Line 251 Galleria delle Stalattiti – this is the first time this name is mentioned – it should appear in the initial description of the field site.

Line 259 – do you see changes in growth rate with $\delta^{13}\text{C}$, since the thickness of the soil is here adduced as a cause of the low ionic strength and therefore slow growth rates?

Line 281 – Late MIS6 – please give an actual date as well for those readers who do not have the dates of all the MIS in their head.

Line 288 produce → reduce

Line 289 em dash → open bracket

Line 290 – “comparing the maximum Dmg value and temperature” – I’m not sure what is meant by this. How is the maximum temperature derived? It seems from the figure caption that the temperature used is 6C, but it’s not clear how this is a maximum or why this temperature has been chosen.

Lines 400-402 – Does this process of recentring the dates for CD3-1 make an assumption that the relative growth rate is invariant between the two speleothems? What are the implications if this turns out not to be the case? This seems to be another source of uncertainty for the age model.

Line 479 100 yr ten then → 100 yr, then

References – references are given in a footnote system rather than alphabetised, but in-text references are name-date rather than footnotes

Fig S1 – I might have misunderstood Fig S1b but the uncertainties given do not seem to match the ones shown by the error bars on Fig S1a?

Fig S6b – Could the very high r here be related to the gap in the middle of the data?

Reviewer #3 (Remarks to the Author):

The use of magnesium (Mg) in calcite speleothems as a recorder of paleo-temperature has been long-sought by the speleothem paleoclimate community. However, in practice the T signal in stalagmite Mg concentrations is typically swamped by variations in water film composition, relating mainly to hydrology. The report of a T-driven variation in Mg in the Corcia sub-aqueous speleothem CD3 is therefore noteworthy if for nothing more than its singular nature (when compared to the published literature). Drysdale et al. report on the close correspondence between Mg in the sub-aqueous CD3 speleothem and marine SST records from the Iberian margin.

For the purposes of this review, it is important to note that I have seen numerous conference reports relating to this record and the Corcia site. I therefore have been somewhat conditioned to the data presented, but need to remind myself of the exceptional nature of the record and findings in the context of the published literature. In this context, the reported findings are indeed

exceptional and point to the huge promise provided by continuously deposited sub-aqueous calcites for reconstructing continental climate change, to provide quantitative paleotemperature records and to anchor marine and ice core records using a combination of wiggle-matching and U-series geochronology. The authors of this paper are well known for their precise, painstaking work and excellence in geochronology. Such expertise is certainly needed for the analysis of such a slow-growing sample. As such I have no technical concerns over the generation of the primary data.

Specific questions and recommendations:

The authors report the positive covariation between Mg and temperature in CD3. This observation is already reported in the Drysdale et al GCA 2019 paper, although the timeseries was not presented. In the current paper the authors provide the Mg timeseries. While certainly new, I found the Mg data to be somewhat frustrating since I was expecting to see a T reconstruction here. The $\delta^{18}O$ to T relation was already shown in Drysdale et al 2004, 2009. Why not take it a step further and reconstruct T using Mg? It would be nice to see the T series compared to any clumped T values you have also obtained. At this point, while you show a strong correspondence to T, you don't demonstrate that Mg can really be used reliably for T reconstruction. In essence, the title is a little misleading since you stop short of using Mg as a paleotemp proxy.

The authors present a remarkable relation between Mg and SST. But how widely is this likely to be seen outside of this specific system? How widespread are such pool deposits? Comment on the potential for changes in pool composition. How much change is needed to overcome the T relation? Is there a way to control for changes in $[Mg]_{aq}$ or do you have to rely on a secondary, independent proxy like D47?

See lines 234-240. Air T fluctuations take time to propagate into the sub-surface due to the role of conduction through the hostrock (see papers by D Dominguez-Villar). How significant are these lags when carrying out your synchronization between isotopes, Mg and ocean sediment records? Since this is more pronounced for deeper caves it seems you may inadvertently miss the lagged response between the external air T and the cave. This seems to be an important factor in assigning/transferring chronologies based on wiggle-matching?

Lines 257-260: I would argue that low supersaturation is likely the main driver of slower precipitation here, not diffusion.

Minor edits:

Missing parenthesis line 162

Delete 'that' line 190

Response to reviewers' comments

We thank the three referees for their very meticulous reviews and thoughtful comments and suggestions. We address each reviewer's concerns individually. Cross references are shown where relevant.

REVIEWER #1

COMMENT 1/1: Most published speleothem records are from sub-aerial stalagmites, but cave pools themselves are not rare (I think that the authors should note this in the text) and may represent a very underutilised resource, with the potential to reconstruct temperature across a wide variety of environments.

RESPONSE 1/1: We have added an extra clause about cave pools in the Introduction where subaqueous speleothems are first mentioned, noting their frequent occurrence. We have also added an extra paragraph in the final section (lines 348-380), where we reiterate this point and list some matters of consideration to clarify the wider potential for using subaqueous calcite speleothem Mg as a T proxy.

COMMENT 1/2: However, I feel that there needs to be some more discussion included before the manuscript could be publishable in Nature Communications. My major comments revolve around the fact that I do not feel that as currently written the manuscript makes a definitive case that temperature controls Mg incorporation into the pool calcite. As I mentioned above, I do agree that this is probably what's happening, but there needs to be more evidence, and more discussion of temperature in the main text. As the authors state on lines 118-119, this may be the first time that Mg in a speleothem has been shown to vary positively with temperature, so this is a potentially controversial result; it therefore needs to be supported carefully. For example, it would be great to emphasise the temperature inside the cave pool and chamber in the main text. There is surprisingly little discussion about the relationship between pool water temperature and outside MAT; I appreciate that there is more discussion in other publications, and some in the methods, but this should be highlighted in the main text. How many temperature measurements were made of the water and of the cave air? Drip water temperature measurements? How stable is the pCO₂ of the cave chamber? Are there no measurements of temperature directly outside the cave entrance? How far away is the Levigliani met station? What is the seasonality like in all these parameters, if present?

RESPONSE 1/2: We entirely agree with the reviewer – there should have been more information on the modern cave conditions to provide some context for the reader. We have now incorporated the cave and external air temperature and cave CO₂ data into the first subsection of the **Results** (lines 106-125) as part of a brief description of the cave setting (moved from the **Methods**). Unfortunately, the CO₂ data are limited to a non-continuous period spanning September 1997 to April 1999. These comprise ~4800 hourly measurements (about 200 days of data) but it does suggest the Galleria delle Stalattiti (GdS) PCO₂ ranges between 1x to 3x the prevailing external atmospheric CO₂

concentrations. Temperatures show no seasonal variation; it is not possible to resolve seasonal variations in CO₂ without additional data.

We have further investigated the controls on cave temperatures. We have inserted an extra paragraph in the final section (lines 311-329), where – following Badino (2018) – we make the case that pool-water and cave-air temperatures approximate the amount-weighted mean annual temperature of the precipitation/infiltration waters at ~1450 m a.s.l. which, as it turns out, is consistent with an earlier, independently derived estimate of the recharge elevation at the cave site.

Note that there is no meteorological station at Levigliani. The nearest met stations are at Retignano (3 km) and Cerviaiole (4.5 km) - see Drysdale et al. (2019).

COMMENT 1/3: Could the cave chamber PCO₂ be different during the last glacial than at present? What effect might this have on pool Mg concentrations, since PCO₂ would affect Prior Calcite Precipitation amounts?

RESPONSE 1/3: Chamber PCO₂ during the last glacial period (and glacials prior to this) cannot be known with precision (as with any cave) but would almost certainly have been lower due to: (i) lower atmospheric PCO₂ (implying a lower PCO₂ of inflowing air during cave ventilation); and (ii) a lower soil PCO₂ due to a cooling-induced dampening of biogenic activity above the cave (which would result in a lower CO₂ budget available for CO₂ outgassing from infiltration waters into the cave air). Both would inhibit PCP, *ceteris paribus*. The cooler percolation waters would further inhibit PCP because the cooler a solution, the more CO₂ it can hold. Glacials are usually drier, which would increase transport times and fracture dewatering, promoting prior calcite precipitation (PCP) (or incongruent dissolution of dolomite (IDD) in the case of dolomitic host rock, which is the case for the gallery in which we are working at Corchia). The tendency for stalagmite CC5 to shift from high to low Mg values across Termination II (new Supp Fig. 8) implies a change from high to low levels of PCP/IDD that exceeds the effect of reduced glacial soil PCO₂ (and percolation water PCO₂) and reduced degassing in the chamber. Thus, the *stalagmite data* indicate enhanced (reduced) PCP/IDD during the glacial (interglacial) state and that hydrological control *significantly exceeds* the temperature effect, i.e. cooler (warmer) temperature would give lower (higher) Mg if it were not for changes in Mg/Ca[*aq*] due to PCP/IDD. We have made this point in the revised ms and included a Supplementary Figure (8).

We are not 100% sure of the point the reviewer is making with regards to the effect of chamber PCO₂ on pool-water Mg concentrations since PCO₂ would affect PCP (degree of Ca²⁺ loss). The pool-water Mg concentration will be a function of the PCO₂ and the residence time of the *percolation waters*, not the PCO₂ of the cave air (unless we've missed something). The higher the percolation-water PCO₂, the more host rock dissolved; and the longer the residence time, the greater the likelihood of: (i) saturation being reached between the host rock and the percolation water; (ii) PCP; and (iii) given the dolomite host rock, the degree of IDD. These three processes will together affect the Mg concentrations (dominated by percolation water PCO₂) and the Mg/Ca (dominated by PCP and IDD, both of which remove Ca to calcite, thus enriching Mg/Ca). The cave air PCO₂ will control the release of CO₂ from the percolation (and pool) waters but not Mg concentrations.

We understand the reviewer's concern that a study arguing for speleothem Mg changes being caused mostly by source-water temperature is controversial, but we suggest this is only the case because cave-pool speleothems, especially those situated in deep cave chambers, have been largely ignored. Clearly, most speleothem scientists just walk past them when caving for speleothem samples. We suggest the community considers these archives as potentially useful information sources in future studies. We have incorporated these arguments into the final paragraph of the main text.

COMMENT 1/4: Another point that should be discussed further is the excess ^{230}Th issue, and whether it will be relevant to other similarly deposited calcites. The authors very correctly identify this as an issue, and it has been previously identified as an issue at Devil's Hole (also noted by the authors), another subaqueous speleothem. Is this likely to be a universal issue with subaqueous speleothems? I don't see any reason why it wouldn't be, and if this is the case, it may not be a simple task to use these sorts of speleothem to correct sea sediment and glacial ice core derived temperature reconstructions via wiggle matching. The chronology here was corrected using wiggle matching to sea sediment records; if the excess Th issue is universal to subaqueous speleothem, can the data be corrected without resorting to synchronisation?

RESPONSE 1/4: Theoretically, the excess ^{230}Th problem should affect all cave-pool speleothems that grow slowly enough, but the issue becomes more extreme as pool-water U content (and therefore supply of ^{230}Th produced in the water column) and pool depth increases, and as calcite growth rate decreases. Moseley et al. (2016) have demonstrated very nicely that corrections are possible in settings like Devil's Hole. They measured U-Th ages on coeval subaqueous speleothem sections (verified by stable isotope profiles) sampled from a range of depths and showed that the age offset increases with speleothem depth, enabling a regression-based age correction. We doubt that a universal, regression-based correction would work in practice, however, because one must also account for past fluctuations in water-table depth and speleothem deposition rates. Wendt et al. (2018) subsequently presented evidence suggesting palaeo water table position can be estimated by dating morphological changes in Devil's Hole speleothems. At Corchia, the context is much different: it is a small and rather shallow cave pool instead of a large regional groundwater system. Water-depth range in Laghetto Basso is 50-100 times lower, so the chances of making a depth-based correction would be highly unlikely. Further, the calcite folia used to mark the position of the water table at Devil's Hole do not exist in Corchia, so a morphology-based estimate of palaeowater levels is not possible, even if a palaeowater-level age correction could be made.

The point we have attempted to make clear in our manuscript is that any proxy information from CD3 that is being used as a tuning target for correlating marine or other archives needs to be anchored to precisely dated, coeval stalagmites (or flowstones). This is best achieved via synchronising their $\delta^{18}\text{O}$ profiles, which allows transfer of the stalagmite chronology to CD3, as we show in Figure 5. The question of whether or not this is a simple task will depend on the confidence one places in the stalagmite-to-subaqueous speleothem correlation, which will vary from cave to cave. In this manuscript, we do not explicitly wish to make claims about the timing of specific palaeoclimate events. However, our ultimate aim is to address these questions in future studies, in which case the synchronisation uncertainties would need to be

conservatively estimated then propagated into any final age estimates. This is a fairly straightforward process (see Tzedakis et al. 2018 and Bajo et al. 2020 for recent examples from Corchia). We agree that not all caves will be appropriate for such investigation. It will be a matter of trial and error. In the light of these points, we have added some text in the final **Implications** section of the paper.

COMMENT 1/5: Line 47: ‘...on G-IG timescales are sparse...’ – tree rings of course are excellent palaeotemperature proxies but on a shorter timescale, so worth quantifying the timescales you are discussing here.

RESPONSE 1/5: Good point - we have made a minor edit here to make it clear we are focusing on multiple G-IG cycles.

COMMENT 1/6: Line 79: other references here: Moseley et al., Science (2016), and Wendt et al., Science Advances (2018).

RESPONSE 1/6: We have added these citations, thank you.

83: ‘concomitant’
Corrected – thank you.

120: no need for quotation marks here
Now removed

COMMENT 1/7: 130-133: It’s good that the authors have taken this into account. The high-resolution data over T-II are a nice inclusion, but with a 0.5 per mil uncertainty in the $\delta^{18}\text{O}$ values, are the SHRIMP values interpretable, other than the large shift at 2.5 mm?

RESPONSE 1/7: The uncertainties on the SHRIMP measurements are indeed large (not atypical given the technique). Nevertheless, in spite of the noise in the raw data, the main features are evidently well-captured, particularly the large shift in the $\delta^{18}\text{O}$ (which we regard as the most critical excursion), as the reviewer observes.

COMMENT 1/8: 234: The cave temperature is not always equal to the outside MAT. Different caves can behave differently, and cold traps can exist when colder (denser) winter air fills a cave, causing cave temperature to be somewhat lower than outside MAT. Again, are any cave air temperature data available for the site? If so, could they be included here? The constant cave air pCO_2 (could the authors list the value) suggests that cave air temperature is similarly constant, as suggested by the authors, but this should be discussed as it is the crux of the manuscript.

RESPONSE 1/8: We have added cave temperature and PCO_2 information - please see Response 1/2 above. We have also added some discussion of what the gallery temperature is likely to mean in terms of external temperatures.

COMMENT 1/9: 256: Could the growth rates be listed here?

RESPONSE 1/9: The mean growth rate of CD3-1 has been removed from the **Methods** and added to the **Study Site** subsection (line 114) and again in the **Discussion** section (line 272) at the point where this reviewer requests.

COMMENT 1/10: 323: “has the potential to record external air temperature changes above the cave.” Again, as mentioned above - are conditions at the cave site comparable to outside conditions? Methods discuss the lapse rate and calculates an outside temperature at the site. This makes sense, but are no direct measurements available? These would obviously be preferable to a calculation. The key point of the manuscript is direct temperature reconstruction using Mg; the key information regarding the site temperature should not be relegated to the Methods – this should be in the main text, and it’d be nice to see more information regarding the local temperature signal, like seasonality.

RESPONSE 1/10: See **Response 1/2**.

Reviewer #2

COMMENT 2/1: As far as I am aware, these claims are novel. The paper implies that claims a and c may be generalised to other subaqueous speleothems, but as yet there have been no studies to verify this. The paper certainly has major implications in this regard, though significant further work will be required to test these claims in other subaqueous speleothems. If it is found that subaqueous speleothems in general accurately record cave temperatures (and therefore mean annual temperatures outside the cave), this will be an exciting step forward for speleothem palaeoclimate, as it will allow well-dated reconstructions of mean annual air temperature in terrestrial regions. As stated in the article, at present such reconstructions are difficult to produce from speleothems as the geochemical signals in speleothem calcite are affected by a range of different climatic and cave parameters.

RESPONSE 2/1: We agree with the statement that “significant further work will be required to test these claims in other subaqueous speleothems”. We have inserted new text at the end of the **Discussion** (lines 348-380) section qualifying the potential of Mg in subaqueous speleothems elsewhere.

COMMENT 2/2: A major comment I have with regard to this paper is its framing of the novelty of its conclusions. The Mg/Ca thermometer is initially validated with reference to $\delta^{18}\text{O}$ and $\delta^{13}\text{C}$ in the same speleothem, as these are interpreted as being related to temperature. Having stated that these temperature proxies already exist, the paper does not explicitly state why the Mg thermometer is of interest.

RESPONSE 2/2: This is a good point – we should have made this much more explicit in our original submission. We have modified the **Results** section to put the case more clearly that Mg has advantages over both $\delta^{18}\text{O}$ and $\delta^{13}\text{C}$ as a temperature proxy. In doing so, we have modified the structure of the **Results** section. We have collapsed the two subsections relating to the CD3 and ocean comparisons over the last 350,000 years into a single subsection and inserted additional text at the end summarising the main points of that section. This includes reference to the fact that, *at orbital scales*, all three CD3 proxies contain a strong temperature component; we qualify this with reference to periods when $\delta^{18}\text{O}$ and $\delta^{13}\text{C}$ are deemed unreliable. We have also collapsed the two original Termination II subsections into one composite section and added further edits that highlight the advantages of Mg over $\delta^{18}\text{O}$. We believe a clearer case is now made for Mg being the main proxy of interest in our paper. We summarise the arguments below.

As this reviewer states, we initially validate the apparent Mg–temperature link on the basis of the strong (negative) covariation between Mg and both $\delta^{13}\text{C}$ and $\delta^{18}\text{O}$ (see r values, Fig. 2f). Our previous work (Drysdale et al. 2004, 2005, 2009, 2012; Zanchetta et al. 2007; Regattieri et al. 2014; Bajo et al. 2017, 2020; Tzedakis et al. 2018) has shown that both $\delta^{13}\text{C}$ and $\delta^{18}\text{O}$ have strong links to temperature. Using this as a foundation stone, the main point of this orbital-scale analysis is to explicitly contrast our own findings with those from many previous speleothem studies that show clear hydrological control of Mg variations. Nevertheless, as we have discussed in our previous work, other factors drive both isotope ratios. For example, although progressive warming associated with the transition from glacial to interglacial conditions results in decreasing speleothem $\delta^{13}\text{C}$, minimum $\delta^{13}\text{C}$ values are attained

well after the thermal optimum of each interglacial. For the case of the Holocene, see Figure 2 in Zanchetta et al. (2007) and the discussion in section 4.5 in Bajo et al. (2017); both studies relate to the same speleothem (CC26) which shows $\delta^{13}\text{C}$ reaching minimum values at around 2-3 ka, about 7 kyr after the SST optimum; for the case of MIS 9, see section 5.2 in Drysdale et al. (2004) relating to speleothem CC1, where the lowest $\delta^{13}\text{C}$ values are reached in MIS 9c, not MIS 9e; and for MIS25 and MIS 21, see the SST versus Corchia speleothem stack $\delta^{13}\text{C}$, which show $\delta^{13}\text{C}$ lagging SSTs for these interglacials (this can be plotted from the data of Bajo et al. (2020) available at <https://www.pangaea.de/?q=Bajo%2C+Petra>). In fact, this lag phenomenon is found for all interglacials of the last million years. Accordingly, this makes $\delta^{13}\text{C}$ less useful than Mg as a temperature proxy.

The $\delta^{18}\text{O}$ in Corchia speleothems is largely driven by changes in rainfall amount. The relatively close alignment between $\delta^{18}\text{O}$ and SSTs (e.g. Drysdale et al. 2004, 2007, 2009) suggests that higher rainfall occurs when regional temperatures are warmer, hence the (indirect) association between temperatures and speleothem $\delta^{18}\text{O}$. However, as Marino et al. (2015) have shown, meltwater discharges during a glacial termination decrease surface-ocean $\delta^{18}\text{O}$ enough to cause the $\delta^{18}\text{O}$ of rainfall reaching the cave to decrease. This corrupts the speleothem $\delta^{18}\text{O}$ -SST link, as vividly shown in the Termination II comparison in our manuscript; Mg during this period (see Fig. 5d) captures the SST increase through T-II beautifully, implying it has superior qualities as a temperature proxy. Further, Mediterranean hydroclimate changes associated with sapropel events can drive the speleothem $\delta^{18}\text{O}$ lower, even when little temperature change has occurred. For example, SSTs were relatively stable between 9 and 7 ka (e.g. Martrat et al. 2014 QSR) yet speleothem $\delta^{18}\text{O}$ decreased – this was due to hydroclimate changes associated with sapropel S1 (Zanchetta et al. 2007). In the revised version of the ms, we highlight the cases of cold sapropels S6 and S8, where cooler but wetter climate produced decreased $\delta^{18}\text{O}$ (more rainfall) as well as lower Mg (cooler conditions), further highlighting Mg as a superior palaeotemperature proxy than $\delta^{18}\text{O}$.

COMMENT 2/3: I assume there is a closer relationship between the Uk37 palaeothermometer and Mg concentrations than there is between Uk37 and either $\delta^{18}\text{O}$ or $\delta^{13}\text{C}$. However, I think that this should be explicitly stated and p-values/correlation coefficients should be produced for Iberian Margin Uk37 vs CD3-1 $\delta^{18}\text{O}$ and $\delta^{13}\text{C}$ (though these could be included in supplementary material). This would help to explain why the speleothem Mg thermometer is of special interest.

RESPONSE 2/3: To elaborate on the above, when we resample the 0–350 ka ocean-core and CD3 data at 5-kyr increments (Fig. 3), the correlation between SST and speleothem $\delta^{18}\text{O}$ is actually stronger (0.79 versus 0.69; it is 0.61 for SST versus $\delta^{13}\text{C}$) than for SST versus speleothem Mg. This supports our previous assertions that speleothem $\delta^{18}\text{O}$ tracks regional SSTs at orbital scales (Drysdale et al. 2004; 2005; 2007). The SST and planktic $\delta^{18}\text{O}$ are themselves highly correlated (0.92), and the speleothem $\delta^{18}\text{O}$ is tuned to the planktic $\delta^{18}\text{O}$ ($r = 0.85$) to get the speleothem data onto the ocean-core time scale for Fig. 3. So, taken together, it is not surprising that the SST and speleothem $\delta^{18}\text{O}$ are highly correlated. The critical point is that we use (for the 0 – 350 ka series in Fig. 3) a resampling interval of 5 kyr. This is constrained by the maximum time increment in the CD3 series. Such a coarse resampling (which is entirely appropriate for looking at orbital-scale relationships between proxies) conceals brief

millennial to sub-millennial intervals of decoupling between SST and planktic that occur (as we allude in the response above) during terminations (see Fig. R1 below, and the following text, where the significance of this is elaborated).

Figure R1: Planktic $\delta^{18}\text{O}$ and SST data from MD01-2443/2444 showing decoupling between the two proxies during the last four terminations (highlighted by the grey panels). This is revealed with higher resolution data but concealed when the series are resampled at 5-kyr increments (as is the case for Fig. 3). For most of the time, both proxies track one another, producing a strong *negative* correlation (see Fig. S3d) because the principal control on both is ocean temperature. However, during terminations, meltwater pulses can briefly decrease both the $\delta^{18}\text{O}$ and SSTs (a *positive* correlation). The advantage of using speleothem Mg over speleothem $\delta^{18}\text{O}$ is that Mg is more likely to track SST changes because the $\delta^{18}\text{O}$ is influenced by many more factors (e.g. rainfall amount, moisture source, storm tracks, temperature).

When we look at Termination II (Fig. 5) with data at a higher resolution (where we use a 500-yr resampling – again, a bin width constrained by the CD3-1 data), we find the correlation between speleothem $\delta^{18}\text{O}$ and SST weaker ($r = 0.62$) compared to that for the 0 – 350 ka series ($r = 0.79$), and is much lower compared to the Mg versus SST ($r = 0.92$). This is due to the speleothem $\delta^{18}\text{O}$ and SST decoupling through the Heinrich 11 meltwater pulse: the surface ocean $\delta^{18}\text{O}$ decrease is transmitted to both the planktic and speleothem $\delta^{18}\text{O}$ at the same time that SSTs decrease (due to the slowdown of the AMOC).

We have added the new scatterplots and r values to a new version of Fig. S3 and clarified the case for Mg as the superior palaeothermometer in the text.

COMMENT 2/4: Towards the end of the paper (line 337), the authors also use the agreement between temperatures reconstructed from clumped isotopes and those reconstructed from Mg/Ca as validation for their argument that Mg/Ca records temperature. Once again, this raises the question of why use Mg/Ca at all if clumped isotopes tell us the same thing. Again, an explicit statement of the utility of Mg/Ca would be useful here.

RESPONSE 2/4: We did not actually make this claim in the text. In the *context of future work*, we stated that a clumped-isotope/Mg calibration should be possible because we have recently reported that clumped-isotope temperatures on modern CD3 calcite agree with modern pool water temperatures. Given that clumped-isotope analysis is *extremely* time consuming (at least 15 replicates to achieve ± 1 deg. C precision on a single data point), and that Mg data can be generated in a fraction of the time, a Mg-based temperature series via a clumped isotope-Mg calibration would be a most efficient way to develop a palaeotemperature series. The utility of the Mg is the ability to measure it accurately, quickly, non-destructively and at very high (micron-level) resolution. We now emphasise these points in the **Discussion** section (lines 336-339). See also **Response 3/4**.

COMMENT 2/5: Furthermore, a major plank of the “Implications” section is that Mg correlations with deep-sea palaeothermometers will allow ocean sediment cores to be placed on a radiometric dating scale. However, both this paper and earlier papers comparing Corchia Cave with ocean cores synchronise ocean sediments with radiometric time-scales by synchronising speleothem $\delta^{18}O$ with various ocean proxies, so it is not clear what advantage Mg gives in this regard. Again, an explicit statement is needed with regard to the utility of Mg concentrations over and above existing methods of correlation.

RESPONSE 2/5: We believe **Responses 2/2** and **2/3** above provide the explanation. The revised manuscript text makes our case much clearer. Having a speleothem property that better captures a temperature signal, especially at a good resolution, is highly valuable – this is clearly recognised by the other reviewers. The conclusions reached by the earlier Corchia papers were based on the data and knowledge at the time. Science progresses, so there should be no surprise that new methods of cave-ocean correlation will emerge as we learn more about the samples, the cave systems and the environmental processes with which we are dealing.

COMMENT 2/6: Finally, I think that a major implication of this paper is that subaqueous speleothems elsewhere might also have highly temperature-dependent Mg correlation coefficients and therefore might also record cave water temperatures (and therefore mean annual air temperatures outside the cave). These air temperatures are of great interest in and of themselves, not merely for the light they can throw on the timing of ocean sediment reconstructions – indeed, well-dated, high-resolution reconstructions of air temperatures in continental interiors would be very exciting. I feel that more could be made of this in the paper, both in the abstract and in the “Implications” section, which would clarify the novelty and the impact of the paper.

RESPONSE 2/6: We tried to make this point clear in the original **Introduction**; we have reinforced this point in the first part of the final paragraph of the **Discussion** section. Note, however, that without an absolute temperature calibration, which is part of ongoing work (and now frustratingly put on hold while covid-19 prevents our group from continuing the clumped-isotope analyses...) we cannot produce the “well-dated, high-resolution reconstructions of air temperatures in continental interiors” that this

reviewer states would be “very exciting” (we are in full agreement here). The main point of this manuscript (and its novelty) is that Mg variations in a speleothem have been shown to act as a fairly robust temperature indicator *for the very first time*.

COMMENT 2/7: In Line 190, the authors state that the d18O and Mg proxies become decoupled at the termination of Heinrich 11 and that this is “critical”, but they do not subsequently discuss the implications of the decoupling of these two proxies. Recharge rainfall d18O is previously identified in the paper as being related to temperature, so some kind of follow-up is needed here to suggest why these two temperature proxies become decoupled at this point and what the climatic reality of this might be. Again, the Mg proxy seems here to be giving us a much more detailed picture of the structure of climate variability at the end of Heinrich 11, so this has implications not just for our understanding of this climate event but for the utility of this proxy for reconstructing climate dynamics in general.

RESPONSE 2/7: We feel we have addressed this point in our previous responses (1/4 and 2/2) and have made the appropriate changes to the revised manuscript.

COMMENT 2/8: The argument about sapropels in lines 154-173 seems like it might have some important implications for the age model used in this section – are there other sapropels during the depositional period and might they have a similar problematic effect on the age model for CD3-1? As far as I understand it, the age model here for CD3-1 is based on the age model for the Iberian Margin cores – how are the dates for S6 and S8 derived? Assuming there is some kind of external date associated with the sapropels, where should these sapropel events be located in terms of the G bulloides record from the IM cores, and do you actually see high d18O in the IM record at these times? If the low Mg and d18O in CD3-1 are assumed to be associated with S6 and S8, wouldn't it be better to use the known dates for these rather than, as the authors state, producing an alignment with incorrect features in the IM G bulloides record?

RESPONSE 2/8: We disagree that this is an issue of age modelling. The ages indicated for the two cold sapropels (grey bands in Fig. 3) are taken off the revised CD3-1 age scale, which, as the reviewer correctly indicates, is based on the age model of IM ocean-core (MD01-2443/2444) with which CD3-1 is being compared. The purpose of this orbital-scale comparison is to determine the extent to which Mg and SST covary in order to support our claim that Mg in CD3-1 is recording temperature.

We introduce the subject of cold sapropels to help explain some of the scatter in the Mg-SST bivariate plot (Fig. 3d). This scatter could be real (the environmental teleconnection temporarily breaks down), or it could be a tuning error (there are periods when it is not appropriate to tune the CD3-1 $\delta^{18}\text{O}$ to the planktic $\delta^{18}\text{O}$). We suggest the latter. Cold sapropels provide a plausible explanation as to why CD3-1 Mg is low (which we suggest is a cold period) at a time when CD3-1 $\delta^{18}\text{O}$ is also low (a wet period, but not a warm period...because Mg is low; normally, CD3-1 Mg and $\delta^{18}\text{O}$ negatively co-vary – see Fig. 2F, middle panel) and SST is high (a warm period). The discrepancy comes about because aligning CD3-1 $\delta^{18}\text{O}$ with the planktic $\delta^{18}\text{O}$ (which is how we apply the IM age model to CD3) is not 100% bombproof because at times of cold sapropels the link between planktic $\delta^{18}\text{O}$ and CD3-1 $\delta^{18}\text{O}$ breaks down. This is

because the planktic $\delta^{18}\text{O}$ *does not* (cannot?) record the pluvial conditions concurrent with this stadial. We have clarified this point in the revised text.

We cannot use the known dates of S6 and S8 because there is no way of tying the occurrence of the sapropels to MD01-2443/2444 because this ocean core does not record these sapropels. (For what it is worth, the ages are ~ 170 - 180 ka for S6 – Bard et al. 2002 EPSL; and ~ 223 - 218 ka for S8 – Bar Matthews et al. 2003 GCA).

COMMENT 2/9: The paper is well-written but it is quite dense. In particular, the various records involved are synchronised to each other in several different ways to validate a series of arguments. Each of these synchronisations involves different levels and sources of uncertainty. I found this synchronisation and resynchronisation quite confusing and it was not clear to me what the implications of the various sources of uncertainty might be for the conclusions presented in the paper. For example, given the high temporal resolution of the reconstruction of the H11 termination, what are the implications of the 4 kyr uncertainty in the IM core age model? Given that one of the major thrusts of the paper concerns comparing records on common timescales, I think it is important to be clear about this aspect. I wonder if a bulleted list or table of synchronisations and their purposes and uncertainties could be included, perhaps in the supplementary material?

RESPONSE 2/9: Synchronisation/age uncertainties are of secondary importance in our paper because we are not making any claims about the *timing* of particular events. If we were, the reviewer would quite rightly have strong case for more quantitative information on uncertainties. We are interested instead in the *phasing* between speleothem and ocean proxies. For the low-resolution comparison (Fig. 2 and 3), we were primarily interested in building a case that CD3-1 Mg is acting as a temperature proxy at the orbital scale by: (i) tracking major changes in CD3-1 $\delta^{18}\text{O}$ and $\delta^{13}\text{C}$; (ii) generally tracking major changes in SST. To do this, we synchronised the CD3-1 $\delta^{18}\text{O}$ to the planktic $\delta^{18}\text{O}$, the scientific basis for which is detailed in the main text. The same logic applies to the high-resolution T-II study, except the chronology of the ocean record has already been anchored to the Corchia *stalagmite* chronology, as reported in Tzedakis et al. (2018). The only new synchronisation here that we bring to the present study is the one between the CD3-1 and stalagmite $\delta^{18}\text{O}$ records: this is to get the CD3-1 data onto an accurate radiometric time scale (given the U-Th dating issues with CD3-1). Having both the ocean record and CD3-1 on the same chronology enables the phasing between CD3-1 Mg and $\delta^{18}\text{O}$ to be compared to that between SST and planktic $\delta^{18}\text{O}$.

We have modified the relevant text in the **Methods** (lines 429-438; and lines 482-495) to improve the clarity of the synchronisation procedures for the both the 0 – 350 ka low-resolution and Termination II high-resolution parts of the study.

COMMENT 2/10: I am not a geochemist, but the question of the partition coefficient for CD3-1 seems to me to be very important. Is it generally known that subaqueous systems have higher/different partition coefficients than subaerial/thin film systems? If so, please state this and give references. If not, what might be the cause of this significant difference in partition coefficients and partition coefficient slopes? This is the key to why CD3-1 Mg responds to temperature rather than hydrology, and therefore the key to

whether this may also happen in other subaqueous speleothems, so it has important implications for the potential impact of this study. I think it is worth a short discussion.

RESPONSE 2/10: Until our article was published last year (Drysdale et al. 2019 GCA), the link between partition coefficients and slow growth rates in subaqueous calcites was unknown because it had not been the subject of study (either in the lab or in a natural setting). Our research showed there are similarities and differences in partition coefficients between CD3 and other calcites according to the element under consideration. Mg happens to have a higher partition coefficient compared to other data for comparable temperatures. We have improved the text in the **Discussion** (lines 291-310) section to emphasise the importance of depositional setting.

There follows a list of more minor comments.

Line 55 Mg/Ca in foram calcite also depends on species
This has been incorporated into the text.

Line 72 around → above?
No, we mean either side of mean values (\pm). 'Around' is in common usage in this context.

Line 83 concomitants → concomitant
Corrected, thank you.

Line 130 the cause of the offset is most likely due to scavenging → the cause of the offset is most likely scavenging
Corrected, thank you

Line 156-158 – black circle – do you mean black square?
It is the back circle, but we forgot to make mention of the black square shown in Fig. 3d. We have rectified this.

Line 165 – intensive → intense
Corrected, thank you

Line 168 – why do both types of sapropel produce reduced d18O?
Both types of sapropel bring more humid conditions. This is recorded due to the rainfall amount effect on speleothem $\delta^{18}\text{O}$. We have clarified this in the text.

Line 169-70 black circle or square? Do these two values represent both S6 and S8 or just S8?
Circle (we have clarified this - please see above). Both sapropels lie within the circle.

Line 214 – I agree the structural agreement is good through the shaded portion, but it is significantly poorer from 125-129 ka. Probably not a big deal given that the argument hinges on H11, but worth being precise.
We have modified the text accordingly.

Line 218 Fig. S5 → Fig. S6

Corrected, thank you

Line 237 – long residence time – quantify this
Information added

Line 238: ~thermal → “approximate thermal”
Correction made

Line 241 patterns consistent with changes in → patterns assumed to be consistent with changes in
Text modified

Line 250 is vital importance → is of vital importance for the response of calcite Mg to cave temperature [or similar]
Text modified

Line 251 Galleria delle Stalatitti – this is the first time this name is mentioned – it should appear in the initial description of the field site.
It has now been introduced earlier – first paragraph of the **Results** section

Line 259 – do you see changes in growth rate with $\delta^{13}\text{C}$, since the thickness of the soil is here adduced as a cause of the low ionic strength and therefore slow growth rates?
We do not have high-resolution ion-probe $\delta^{13}\text{C}$ data from CD3 to test this in the current study (the SHRIMP-SI only measures $\delta^{18}\text{O}$), but our previous work has shown that periods of lower $\delta^{13}\text{C}$ are generally associated with higher growth rates and *vice versa*.

Line 281 – Late MIS6 – please give an actual date as well for those readers who do not have the dates of all the MIS in their head.
Text modified

Line 288 produce → reduce
Changed

Line 289 em dash → open bracket
Changed

Line 290 – “comparing the maximum Dmg value and temperature” – I’m not sure what is meant by this. How is the maximum temperature derived? It seems from the figure caption that the temperature used is 6C, but it’s not clear how this is a maximum or why this temperature has been chosen.
It should have read ‘glacial’, not maximum. We have modified the text to make this clearer.

COMMENT 2/11: Lines 400-402 – Does this process of recentring the dates for CD3-1 make an assumption that the relative growth rate is invariant between the two speleothems? What are the implications if this turns out not to be the case? This seems to be another source of uncertainty for the age model.

RESPONSE 2/11: The reviewer is correct to raise this point. Yes, we do make the assumption of relatively constant growth rates, but we are not dealing with two separate speleothems. Both cores were drilled from different positions from the *same* speleothem – a hemispheric, dome-shaped concretion. The same sequence of growth patterns is evident in the two sections. In any case, the implications of variable relative growth rates are minor: there is no weighting attributed to the accuracy of the CD3 chronology besides highlighting (and due to) the age-offset (i.e. Th scavenging) issue. Further, the sampling error (± 1 mm) is extremely conservative (given that we sampled using an automated drilling system to shallow depth), and any minor depth offset would be captured in the sampling error, which is propagated into the age model in any case.

If the reviewer is suggesting that the scavenging issue is actually an artefact of the mis-assignment of the CD3-1 dating positions to the CD3-2 depth scale, we can confirm that the age offset issue is real, as it has been discussed in Drysdale et al. (2019) for the Holocene. It is also clearly evident in the results of unpublished graduate thesis research by one of the co-authors (E. Corrick), which comprises a continuous time series from one CD3 section from which stable isotope measurements and U-Th ages were derived from each powdered sample. When the isotope time series are compared to stalagmite data from the same cave (the stalagmite ages are unaffected by Th scavenging), the age offset is abundantly clear.

Line 479 100 yr ten then → 100 yr, then
Changed – thank you.

References – references are given in a footnote system rather than alphabetised, but in-text references are name-date rather than footnotes

Nature Communications guidelines stress that it is not necessary to use the final formatting system at the initial submission stage. We have changed the formatting for this revised version.

Fig S1 – I might have misunderstood Fig S1b but the uncertainties given do not seem to match the ones shown by the error bars on Fig S1a?

The age errors in Fig. S1b are derived from the age-model output and are thus the same as the errors shown in Fig. S1a. Note, however, that age uncertainties in Fig. S1a should be read using the *y*-axis (i.e. they extend in the *vertical* dimension of the shaded uncertainty envelope, *not* in the horizontal dimension). We hope this clarifies things.

Fig S6b – Could the very high *r* here be related to the gap in the middle of the data?

The gap in the middle is unavoidable because the transition showing the rise in SSTs and cave temperatures is rapid and we resampled both series at 500-yr increments (this interval itself being constrained by the minimum resolution of the data series) to calculate the correlation coefficient, which requires the same number of *x* and *y* observations (i.e. *x-y* pairs).

Reviewer #3

COMMENT 3.1: For the purposes of this review, it is important to note that I have seen numerous conference reports relating to this record and the Corchia site. I therefore have been somewhat conditioned to the data presented, but need to remind myself of the exceptional nature of the record and findings in the context of the published literature. In this context, the reported findings are indeed exceptional and point to the huge promise provided by continuously deposited sub-aqueous calcite s for reconstructing continental climate change, to provide quantitative paleotemperature records and to anchor marine and ice core records using a combination of wiggle-matching and U-series geochronology. The authors of this paper are well known for their precise, painstaking work and excellence in geochronology. Such expertise is certainly needed for the analysis of such a slow-growing sample. As such I have no technical concerns over the generation of the primary data.

RESPONSE 3.1: We thank this reviewer for their generous praise of our work.

COMMENT 3/2: The authors report the positive covariation between Mg and temperature in CD3. This observation is already reported in the Drysdale et al GCA 2019 paper although the timeseries was not presented.

RESPONSE 3/2: The GCA paper that the reviewer mentions is actually concerned with modern partitioning between Laghetto Basso pool waters and CD3 calcite for a range of trace elements. The paper is a natural precursor to the present manuscript because it lays out the modern hydrogeochemical setting. The only reference in the GCA paper to links between Mg and temperature is where we compare the partitioning coefficient for Mg (D_{Mg}) with the D_{Mg} derived from other (non-Corchia) studies *under a range of temperature conditions* (Table 10 and Figure 5 of that paper). The actual paper does not mention a positive covariation between Mg and temperature in CD3 because we only considered the modern pool water and calcite. It is possible that this reviewer has seen some of the data presented in the current manuscript at a recent conference, e.g. AGU Fall Meeting December 2019?

COMMENT 3/3: In the current paper the authors provide the Mg timeseries. While certainly new, I find the Mg data to be somewhat frustrating since I was expecting to see a T reconstruction here. The d18O to T relation was already shown in Drysdale et al 2004, 2009. Why not take it a step further and reconstruct T using Mg?

RESPONSE 3/3: We cannot produce an independent temperature reconstruction because, as the data currently stand, we do not have a robust Mg-T calibration function (although it is being worked on – please see next response). All we have currently is an estimate of the calcite Mg value at the modern pool temperature and Mg time series presented in Figures 3 to 5. A Mg-temperature transfer function *could* be derived by using the modern Mg concentration and modern cave temperature (~8°C) and anchoring the Last Glacial Maximum Mg to the temperature difference between modern and LGM SSTs. However, this would not be an independently derived calibration because it assumes the modern-LGM SST temperature range (i.e. interglacial minus

glacial T) is identical to that of the cave. We prefer to hold off until we have developed the calibration curve. We allude to how we might proceed in the **Implications** part of the revised ms.

COMMENT 3/4: It would be nice to see the T series compared to any clumped T values you have also obtained.

RESPONSE 3/4: The reviewer makes a fair comment about the use of clumped isotopes and CD3. Our work on Mg vs clumped isotopes is the subject of ongoing research by one of our PhD students (who is not a listed co-author), and a robust Mg- $\Delta 47$ temperature calibration curve far from completion. A robust calibration requires the testing of the stationarity of the Mg-clumped isotope function, and a solid quantification of the variability of elemental Mg across individual growth layers (we have quantified modern variability, but cross-layer variability may itself vary with climate).

[REDACTED]

We trust that this satisfies any doubts that the reviewer (and editor) may have about the plausibility of Mg being a palaeothermometer in CD3, and that by not including these data does not detract from the novelty of our work, which has been pointed out by all three reviewers.

[REDACTED]

COMMENT 3/5: At this point, while you show a strong correspondence to T, you don't demonstrate that Mg can really be used reliably for T reconstruction. In essence, the title is a little misleading since you stop short of using Mg as a paleotemp proxy.

RESPONSE 3/5: We have changed the title slightly to avoid any inadvertent misleading of the paper's future readership, whilst maintaining the paper's principal message.

COMMENT 3/6: The authors present a remarkable relation between Mg and SST. But how widely is this likely to be seen outside of this specific system? How widespread are such pool deposits? Comment on the potential for changes in pool composition. How much change is needed to overcome the T relation? Is there a way to control for changes in [Mg]_{aq} or do you have to rely on a secondary, independent proxy like D47?

RESPONSE 3/6: Again, all good questions here. Regarding the applicability of the Mg-SST relationship outside of the Corchia system, this is really a matter for the speleothem community to investigate because virtually all of the climate-reconstruction focus has been based on the interrogation of stalagmites. Following the presentation of our work at AGU in December 2019, the lead author (RD) was approached by several senior (U.S.) colleagues who were unaware of the prospects of subaqueous speleothems and at the same time very excited by the potential opportunities offered by the cave systems that they are currently studying. We would expect a surge of interest following publication of our manuscript. For the record, we have recently recovered subaqueous speleothems from a central Australian cave system whose Mg also appears to preserve a pool-water temperature record show step-like decreases in Mg concentration across several intervals of the Last Glacial period. There is excellent potential – we just need to look beyond stalagmites.

We are not sure how much change in Mg/Ca_(aq) is needed to overcome the T relationship, but, as we allude to in the manuscript, we suspect the potential of Mg as a T recorder is as much to with the kinetics of slow subaqueous calcite growth than a low range of Mg/Ca_(aq). We note in our manuscript that stalagmite CC5 shows a classical hydrological response across Termination II in its Mg, in spite of it being fed by drip waters of similar composition to Laghetto Basso (the source waters of CD3). NB: CC5 was collected next to, and its drip waters would have drained to, Laghetto Basso.

We have yet to find a way for controlling for Mg/Ca_(aq) variability but the search continues. At this stage, D47 appears the best way to cross check and constrain the Mg-T relationship.

COMMENT 3/7: See lines 234-240. Air T fluctuations take time to propagate into the sub-surface due to the role of conduction through the hostrock (see papers by D Dominguez-Villar). How significant are these lags when carrying out your synchronization between isotopes, Mg and ocean sediment records? Since this is more pronounced for deeper caves it seems you may inadvertently miss the lagged response between the external air T and the cave. This seems to be an important factor is assigning/transferring chronologies based on wiggle-matching?

RESPONSE 3/7: The very interesting Dominguez Villars et al. (2014) study considers Postojna Cave, whose air flow is related to external temperature changes owing to its very large entrances. Corchia Cave by contrast has numerous small entrances and is less susceptible to the processes described by Dominguez Villars et al. (2014). These conditions render the mean temperature of its interior galleries most influenced by the mean temperature of the infiltration waters reaching those galleries (please see our arguments on lines 311-329). Given the different controls on the cave air and water temperatures, we do not consider time lags to be an issue.

COMMENT 3/8: Lines 257-260: I would argue that low supersaturation is likely the main driver of slower precipitation here, not diffusion.

RESPONSE 3/8: We have modified the text here to clarify our arguments.

Minor edits:

Missing parenthesis line 162

We have moved note about sapropel ages to the caption of Figure 3.

Delete 'that' line 190

Removed, thank you.

REVIEWERS' COMMENTS:

Reviewer #1 (Remarks to the Author):

The authors have done a very nice job of considering and incorporating (where relevant) my comments from my previous review. The manuscript reads very well and is now very comprehensive, and I do not require any more changes; I recommend accepting without further revision. Well done to the authors - this work is likely to make quite an impact.

Reviewer #2 (Remarks to the Author):

I have read the revised manuscript and the rebuttal letter, and I would like to commend the authors on the level of care they have put into both modifying the manuscript in many cases and explaining their reasoning where they do not consider this to be necessary. The responses to my own comments were clear and detailed, and I am satisfied that my concerns have been addressed. The MS reads well and I think the novelty of the work will be clear to both specialists and those in adjacent fields. I look forward to seeing this published in Nature Communications.

I have a few minor comments, as follows:

Line 13 tracks → track

Line 96 – deposition setting → depositional setting

Line 189-194 – “The aligning of low planktic $\delta^{18}\text{O}$ and low speleothem $\delta^{18}\text{O}$ during cold sapropels” is followed by the statement that planktic $\delta^{18}\text{O}$ does not decrease and is relatively high during these events. I found this confusing, as it wasn't clear to me whether the authors were saying that the planktic $\delta^{18}\text{O}$ was low or high during cold sapropels.

Line 446 I think this should be Sfig 4, not 3?

Figs 5a and S6 – not clear why the dataset which is referred to as CCSS-18 in the figure captions and text is labelled as CC5convd18O in these figures

Line 279 – that either → either that

Lines 304-308 – this seems to cite the high glacial Mg/Ca in CC5 twice?

Line 385 – “therefore” would be better as “however”

Line 495 – Supplementary Figure 5 → Supplementary Figure 6

Line 764 – in the figure caption for Figure 3b, explain the significance of the black arrows.

Response to comments from Reviewer #2

Line 13 tracks → track
changed

Line 96 – deposition setting → depositional setting
changed

Line 189-194 – “The aligning of low planktic $\delta^{18}O$ and low speleothem $\delta^{18}O$ during cold sapropels” is followed by the statement that planktic $\delta^{18}O$ does not decrease and is relatively high during these events. I found this confusing, as it wasn't clear to me whether the authors were saying that the planktic $\delta^{18}O$ was low or high during cold sapropels.

Rewritten as: *However, the Iberian margin planktic $\delta^{18}O$ remains relatively high during these intervals due to surface-ocean cooling at a time of stadial conditions. This highlights an inconsistency in aligning the Corchia $\delta^{18}O$ and Iberian Margin planktic $\delta^{18}O$ during cold sapropels (particularly S8) because these events do not appear to affect the isotopic composition of the surface ocean waters of the Iberian margin³⁹.*

Line 446 I think this should be Sfig 4, not 3?
Correct - changed

Figs 5a and S6 – not clear why the dataset which is referred to as CCSS-18 in the figure captions and text is labelled as CC5convd180 in these figures
We have renamed the CC5convd180 to CCSS-18. We erred here: this part of the Corchia Cave stack from Tzedakis et al. (2018) is mostly, but not entirely, from stalagmite CC5. It is more correct to refer to this time series as CCSS-18. We have changed the axis labels in both figures accordingly.

Line 279 – that either → either that
Changed

Lines 304-308 – this seems to cite the high glacial Mg/Ca in CC5 twice?
We have removed the sentence *“This conclusion is supported by the stalagmite CC5 results for T-II⁴⁹, which point to higher glacial Mg/Ca_[aq] compared to interglacial values.”*
We have revised the text as follows (from line 299): *“The evidence from stalagmite CC5 supporting hydrologically driven changes in Mg/Ca_[aq] (Supplementary Figure 7) suggests it is highly unlikely that pool-water Mg/Ca_[aq] would be equivalent to or lower than modern values due to glacial climates at Corchia being drier than interglacials^{43,49}. We can therefore assume that a D_{Mg} of 0.029 is the maximum possible value.”*

Line 385 – “therefore” would be better as “however”
Changed

Line 495 – Supplementary Figure 5 → Supplementary Figure 6
All figures numbers have been checked and corrected .

Line 764 – in the figure caption for Figure 3b, explain the significance of the black arrows.

The penultimate sentence of the caption reads: *“Black arrows in (B) indicate intervals where exceptionally low SSTs are not matched by correspondingly low Mg concentrations.”*